# Reconstitution of kinetochore motility and microtubule dynamics reveals a role for a kinesin-8 in establishing end-on attachments

**Julia R Torvi[1,2†], Jonathan Wong[1†], Daniel Serwas[1], Amir Moayed[1], David G Drubin[1,2], Georjana Barnes[1*]**

[1]Department of Molecular and Cell Biology, University of California, Berkeley, Berkeley, United States; [2]Biophysics Graduate Group, University of California, Berkeley, Berkeley, United States

**\*For correspondence:**
gbarnes@berkeley.edu

[†]These authors contributed equally to this work

**Competing interest:** The authors declare that no competing interests exist.

**Abstract** During mitosis, individual microtubules make attachments to chromosomes via a specialized protein complex called the kinetochore to faithfully segregate the chromosomes to daughter cells. Translocation of kinetochores on the lateral surface of the microtubule has been proposed to contribute to high fidelity chromosome capture and alignment at the mitotic midzone, but has been difficult to observe in vivo because of spatial and temporal constraints. To overcome these barriers, we used total internal reflection fluorescence (TIRF) microscopy to track the interactions between microtubules, kinetochore proteins, and other microtubule-associated proteins in lysates from metaphase-arrested *Saccharomyces cerevisiae*. TIRF microscopy and cryo-correlative light microscopy and electron tomography indicated that we successfully reconstituted interactions between intact kinetochores and microtubules. These kinetochores translocate on the lateral microtubule surface toward the microtubule plus end and transition to end-on attachment, whereupon microtubule depolymerization commences. The directional kinetochore movement is dependent on the highly processive kinesin-8, Kip3. We propose that Kip3 facilitates stable kinetochore attachment to microtubule plus ends through its abilities to move the kinetochore laterally on the surface of the microtubule and to regulate microtubule plus end dynamics.

## Editor's evaluation

Kinetochores are large protein complexes that mediate faithful chromosome segregation in eukaryotes. The authors develop an in vitro approach to study interactions between kinetochores and microtubules in yeast cell extracts. They use this powerful lysate-based system to characterize a new role for the budding yeast kinesin-8, Kip3, in powering lateral kinetochore movement along microtubules. This paper should be of interest to researchers working in the field of mitosis, the cell cycle, and the cytoskeleton, and, more broadly, for those studying macromolecular complexes with reconstitution and in vitro imaging approaches.

## Introduction

During mitosis, a cell's replicated chromosomes are equally segregated to two daughter cells. In order for this process to happen accurately and efficiently, the microtubule cytoskeleton must completely remodel to form a bipolar spindle. Replicated chromosomes then attach to microtubules, emanating from opposite spindle poles, through a large dynamic chromosome-associated protein complex called

the kinetochore. Proper kinetochore-microtubule attachment is vital for preservation of genomic integrity and prevention of cancer and birth defects (*Weaver and Cleveland, 2005*). Therefore, kinetochore attachment is highly regulated and tightly controlled (*Tanaka, 2013*).

During establishment of biorientation, kinetochores have been observed to first attach to the lateral surface of a microtubule (*Tanaka, 2012*). Such kinetochores then must be transported along the microtubule lattice, where they transition to an appropriate, and stable, end-on attachment (*Rieder and Alexander, 1990*; *Tanaka et al., 2005a*; *Kapoor et al., 2006*). It is only after each sister kinetochore makes a proper end-on attachment with microtubules from opposite spindle poles that the cell will progress into anaphase and complete mitosis. This lateral to end-on attachment conversion has been observed in several eukaryotic species (*Magidson et al., 2015*; *Maiato et al., 2017*), but the underlying mechanisms have been elusive due both to the difficulty of observing this event in vivo and to biochemical hurdles in vitro.

Establishment of end-on kinetochore attachments to microtubules, for chromosomes that are initially attached to spindle microtubules laterally, can be divided into two processes: (1) chromosome lateral movement along the microtubule surface and (2) conversion from lateral to end-on attachment. With regard to the kinetochore reaching the microtubule tip, CENP-E and chromokinesins are particularly important for chromosome movement to the spindle midzone in metazoans (*Craske and Welburn, 2020*; *Lemura and Tanaka, 2015*). However, fungi do not appear to have kinesin-7 or chromokinesin orthologs. Lateral to end-on conversion has been observed in fungi, but these events required that a depolymerizing microtubule tip meet the laterally bound kinetochore (*Kalantzaki et al., 2015*). Other studies in fungi have also implicated minus-end directed motor-based transport of unattached kinetochores (*Tanaka et al., 2005a*), similar to a proposed metazoan mechanism involving the motor protein dynein and the mitotic spindle protein NuMA (*Kapoor et al., 2006*; *Li et al., 2007*).

One class of proteins that might be involved in lateral kinetochore movements is the kinesin-8 motor proteins. This family of proteins includes key regulators of kinetochore-microtubule dynamics. Through a dual-mode mechanism of destabilizing growing microtubules and stabilizing depolymerizing microtubules, kinesin-8 proteins have been proposed to tune microtubule dynamics for proper chromosome segregation (*Risteski et al., 2021*). In budding yeast, Kip3, a highly processive kinesin-8, is required for kinetochore clustering into two discrete foci at metaphase, but how this activity relates to establishment of end-on attachments is not clear (*Wargacki et al., 2010*; *Su et al., 2011*). Declustered kinetochores may represent misaligned or defectively attached chromosomes (*Wargacki et al., 2010*; *Edzuka and Goshima, 2019*; *Shimogawa et al., 2010*). Additionally, in yeast that lack Kip3, chromosomes lag behind the main chromosome mass during anaphase (*Tytell and Sorger, 2006*). The metazoan kinesin-8, Kif18a, is responsible for tuning kinetochore-microtubule dynamics, which ultimately affects chromosome alignment and equal segregation into the daughter cells (*Stumpff et al., 2008*; *Stumpff et al., 2012*). Lagging chromosomes are common in cells lacking Kif18a and result in micronuclei formation (*Fonseca et al., 2019*). Additionally, the *Drosophila* kinesin-8, Klp67A, shares many of the roles of other kinesin-8 proteins including stabilizing and destabilizing microtubules, but this kinesin also stabilizes kinetochore-microtubule attachments (*Edzuka and Goshima, 2019*). Taken together, these observations indicate that kinesin-8 motor proteins have roles both in regulating microtubule dynamics and in establishing kinetochore-microtubule attachments, while potentially playing a role in the formation of proper end-on attachments.

Currently, methods to observe single kinetochores transported along single microtubules are limited. An ideal assay would enable tracking of single intact kinetochores on microtubules assembled from tubulin isolated from the same species as those kinetochores; ideally a species with robust genetics so mutants could be used, and an assay that utilizes the full complexity of a cell. A cell-free system combines the full biochemical complexity of a cell with the experimental control provided by in vitro experiments. In this study, we investigated mitotic regulation of kinetochore-microtubule attachments and dynamics using a reconstitution system that employs cell cycle arrested extracts from the genetically tractable organism *Saccharomyces cerevisiae* (*Bergman et al., 2018*).

## Results

### Reconstitution of dynamic kinetochores on single yeast microtubules

To investigate the mechanism of kinetochore-microtubule attachment at metaphase, we sought to visualize single kinetochores on single microtubules. For these studies, we employed a reconstitution assay wherein cell-cycle arrested whole cell yeast lysate was used to study regulation of microtubule dynamics (*Bergman et al., 2018*). Using this assay, we asked whether kinetochores and their dynamics could be observed by TIRF microscopy (*Figure 1a*). Fluorescently tagged kinetochore proteins from each of the major kinetochore sub-complexes were expressed at endogenous levels in strain backgrounds that also expressed fluorescently tagged alpha-tubulin (Tub1) and a temperature-sensitive allele (*cdc23-1*) of the gene that encodes a subunit of the anaphase-promoting complex, to arrest cells in metaphase at the non-permissive temperature (*Irniger et al., 1995*). The proteins we chose to tag spanned the entirety of the kinetochore (*Figure 1b*).

### Outer kinetochore proteins are dynamic on microtubules

To begin, we tagged two outer kinetochore proteins known to bind to microtubules, Ndc80 (*Cheeseman et al., 2006*; *Wei et al., 2007*) and the Ask1 subunit of the Dam1 complex (*Miranda et al., 2005*; *Westermann et al., 2005*; *Jenni and Harrison, 2018*). Ndc80 and Ask1 kinetochore signals appeared on the microtubule surface (*Figure 1c*) and moved processively at an average rate of 0.56 µm/min toward the plus end of the microtubule (*Figure 1d*). Upon reaching the tip of the microtubule, Ndc80 and Ask1 appeared to transition irreversibly from lateral to end-on attachment. This end-on binding was then followed by microtubule depolymerization where Ndc80 and Ask1 tracked the depolymerizing microtubule end. The mean microtubule depolymerization rate also slowed from 0.82 µm/min (*Bergman et al., 2018*) to an average of 0.4 µm/min upon kinetochore binding to the microtubule tip (*Figure 1d*). After depolymerization onset, the microtubule was never observed to resume growth. Additionally, kinetochore proteins rarely, if ever, detached from the microtubule, either while bound laterally or to the microtubule tip. The directional switch of bound kinetochores was striking, as reflected in the dynamics of these kinetochore proteins and the percent of time the associated microtubules spent in the growing versus shrinking phase (*Figure 1e*). While laterally bound, kinetochores only move towards the plus end of the microtubule. When the kinetochore transitions from lateral to end-on attachment, the kinetochore only tracks the depolymerizing end of the microtubule and thus moves toward the minus-end of the microtubule.

In addition to the Ask1 movement that is the same as the Ndc80 movement: laterally along the MT toward the plus end mentioned above, we also observed a distinct Ask1 population that showed a unique behavior compared to other proteins assayed (*Figure 1c*). This Ask1 population was characterized by a very low intensity and diffuse signal, which briefly associated with and moved randomly along the microtubule. The signal to noise ratio was very low and it moved faster than our sampling rate. For these reasons, we did not investigate this Ask1 signal further. However, this observation potentially provides insight into Dam1 complex ring assembly and function on the microtubule (*Gestaut et al., 2008*; *Grishchuk et al., 2008*).

### Inner kinetochore proteins are also dynamic on microtubules

We next tagged and assayed additional kinetochore proteins from the inner and middle kinetochore regions to determine their behaviors and to begin to assess whether all the kinetochore proteins we observed associated with microtubules as intact kinetochore-like complexes. These additional labeled proteins included: Spc105, Mtw1 (MIND complex) Cnn1, Okp1 (COMA complex), Mif2, Ctf3 (CBF3 complex), and Cse4 (a centromeric histone variant) *Figure 1* (*Westermann et al., 2007*; *Biggins et al., 2013*). Strikingly, for each kinetochore protein analyzed, we observed dynamics similar to those we observed for Ask1 and Ndc80 (*Figure 1—video 1*). Two proteins, Mif2 and Cse4, moved at only 67–75% of the velocity of the others on the microtubule lattice. One possible explanation for the slower velocities of these two proteins is that their GFP protein tags might affect their function; but since these two proteins colocalize with other kinetochore proteins in lysate from dual-labeled strains (*Figure 2*), we conclude that all the proteins examined are present in common complexes. Overall, these seven middle and inner kinetochore proteins have no known microtubule binding capabilities, implying that the kinetochore signals that we observed might represent intact or nearly intact kinetochores on microtubules in our assay.

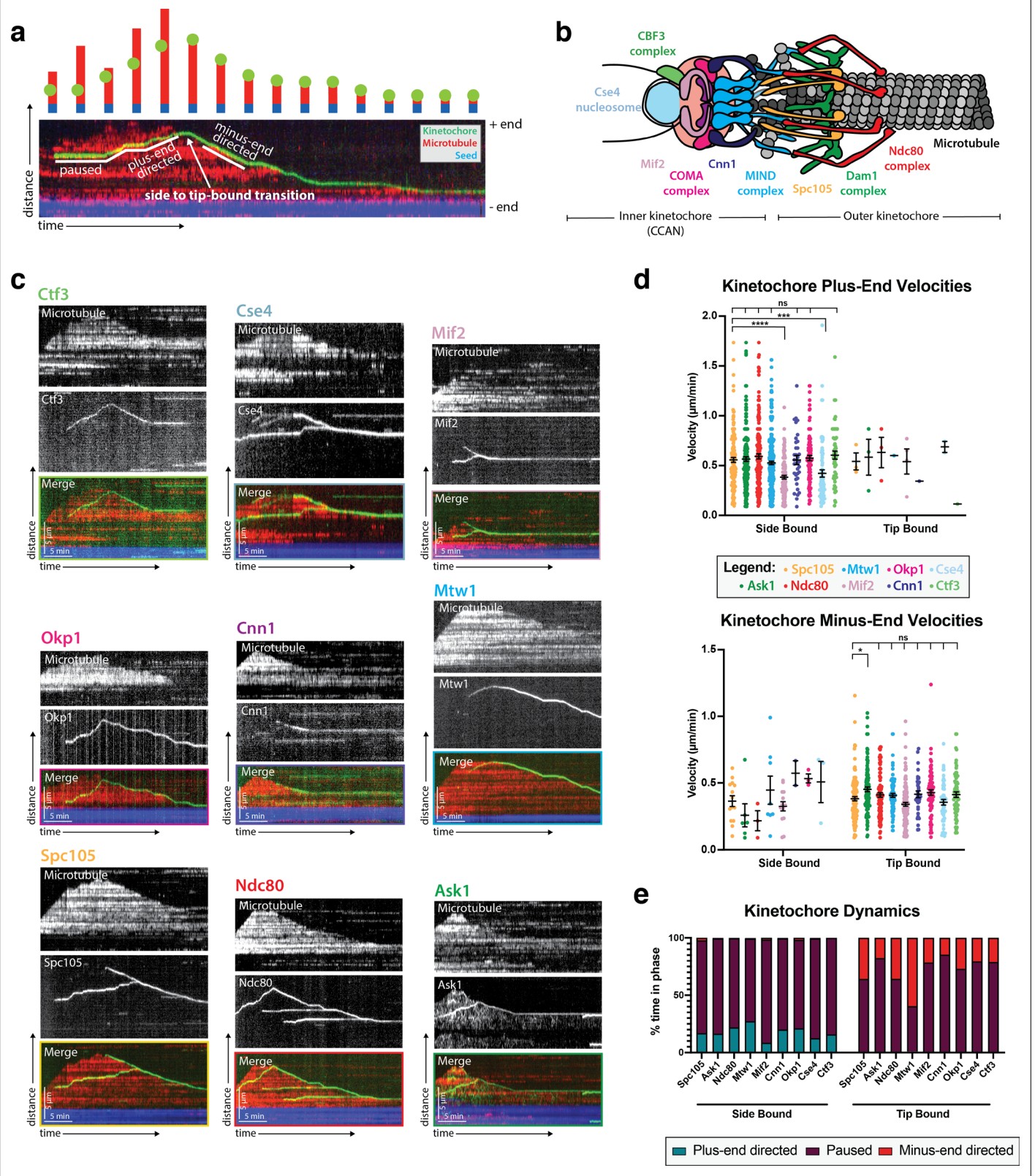

**Figure 1.** Kinetochore reconstitution and dynamics on the microtubule (**a**) Schematic of a typical kymograph representing a dynamic kinetochore on a microtubule. Time (min) is on the x-axis and distance (μm) is on the y-axis. Above the kymograph is a cartoon depiction explaining how the kymograph shows three-dimensional data in two-dimensional space. In both the kymograph and cartoon, the yeast microtubule is red, the kinetochore is green, and the porcine microtubule seed is blue. (**b**) A schematic of the budding yeast kinetochore with its major sub-complexes shown in different colors.

*Figure 1 continued on next page*

*Figure 1 continued*

These colors correspond to the different kinetochore proteins assayed in (**c**). The kinetochore is composed of inner kinetochore proteins such as the Cse4 nucleosome and the CBF3 complex, that bind to DNA, and outer kinetochore proteins such as the Ndc80 and Dam1 complex, that bind to the microtubule. (**c**) Reconstituted kinetochore proteins associated with yeast microtubules. Representative kymographs of kinetochore proteins from each sub-complex are shown. Time is on the x-axis and distance is on the y-axis (scale bars are 5 min and 5 µm, respectively). Kinetochore proteins are green and yeast microtubules are red. The porcine seed is blue. Each of the proteins assayed displays similar dynamics: moving toward the microtubule plus end when bound to the microtubule lateral surface. When the kinetochore proteins reach the plus end of the microtubule, they appear to convert from lateral to end-on attachment, which is often followed by microtubule depolymerization with the kinetochore remaining associated with the plus end. The microtubules were never observed to grow again following the depolymerization events in which a kinetochore protein was attached end-on. (**d**) Quantification of velocities and dynamics of kinetochores bound to microtubules. Kinetochores bound to the sides of microtubules move toward the microtubule plus end at an average velocity of 0.56 µm/min. There were negligible minus-end directed movements when the kinetochore was side bound. When the kinetochores were associated with the microtubule tips, they tracked the microtubule tips as they depolymerized, moving at an average velocity of 0.4 µm/min. There were also negligible plus-end directed movements when the kinetochore was bound to the tip. Each data point in the graph is the average velocity of one kinetochore. The error bars are the mean and standard error of the mean from four replicate trials of two biological replicates. Statistical analysis was done by a Kruskal-Wallis test where **** is p<0.0001, *** is p=0.0003, and * is p=0.0132. For the different kinetochore proteins: from Spc105 in order in the figure to Ctf3, N=128, 155, 116, 102, 174, 56, 77, 59, 87 proteins tracked, respectively. (**e**) When the kinetochore is bound to the side of a microtubule, about 30% of its time it is in plus-end directed motion and the remaining 70% of the time it is paused (not moving). Kinetochores associated with plus ends spend 40% of the time moving toward the minus-end and 60% of the time paused.

The online version of this article includes the following video, source data, and figure supplement(s) for figure 1:

**Source data 1.** Source data supplied.

**Figure supplement 1.** Kinetochore motility is specific to metaphase extracts.

**Figure 1—video 1.** Dynamic kinetochores on the microtubule Spc105 (green) moves on the lattice of the microtubule (red) toward the plus-end.
https://elifesciences.org/articles/78450/figures#fig1video1

Importantly, the kinetochore protein dynamics observed in metaphase-arrested cell lysates were specific to metaphase. When Spc105 dynamics were assayed in S-phase (*cdc7-1*) or anaphase (*cdc15-2*) arrested extracts (*Goranov et al., 2009*), the kinetochore proteins were stationary and did not move on the microtubule lattice (*Figure 1—figure supplement 1a-c*). Additionally, when Spc105 was observed at the microtubule tip, no tip-associated microtubule depolymerization occurred. Therefore, the kinetochore dynamics of the proteins shown in *Figure 1* are specific to metaphase.

## Kinetochore proteins from different sub-complexes move together on microtubules

The observation that inner kinetochore proteins (such as Cse4) which do not physically bind to microtubules, are in fact able to bind to and translocate on microtubules in our assay provided evidence that these kinetochore proteins were not alone or in small sub-complexes, but, in fact, were part of complete or nearly complete kinetochores. To explore this possibility further, we turned to dual-color imaging of kinetochore protein pairs and colocalization analyses. Due to its well-documented and canonical microtubule binding abilities, Ndc80 was chosen as a fiducial marker to which the localization of other kinetochore proteins was compared (*Ciferri et al., 2007*). One possibility was that proteins that reside further away from the Ndc80 complex would colocalize less frequently in our assay than proteins located closer to or adjacent to the Ndc80 complex (*Figure 1b*). On the contrary, when imaging different kinetochore proteins together with Ndc80, we, in fact, saw colocalization independent of that kinetochore protein's proximity to Ndc80 within the kinetochore (*Figure 2—video 1*). Additionally, when using Mtw1, a more 'central' kinetochore protein, as a fiducial marker, we still observed colocalization between kinetochore proteins (*Figure 2*). These data indicate that the characterized movements of single tagged kinetochore proteins in this assay likely represents entire kinetochore complexes.

In our assay, the mScarlet-I fluorophore used to tag proteins blinked on and off (*Klementieva et al., 2017*). Other red fluorescent protein tags tested, including mScarlet, TagRFP-T, and mRuby2, also blinked and/or bleached rapidly. From our single protein tagging (with GFP) experiments (*Figure 1*), we knew that Ndc80 and Mtw1 never disassociated from microtubules to which they were bound. Therefore, the patchy signal we saw with mScarlet-I fluorophore (*Figure 2a*) was likely a result of our red fluorescent protein blinking on and off. This also meant that it was nearly impossible to draw any conclusions about kinetochore assembly on the microtubule at any single point in time (i.e. during

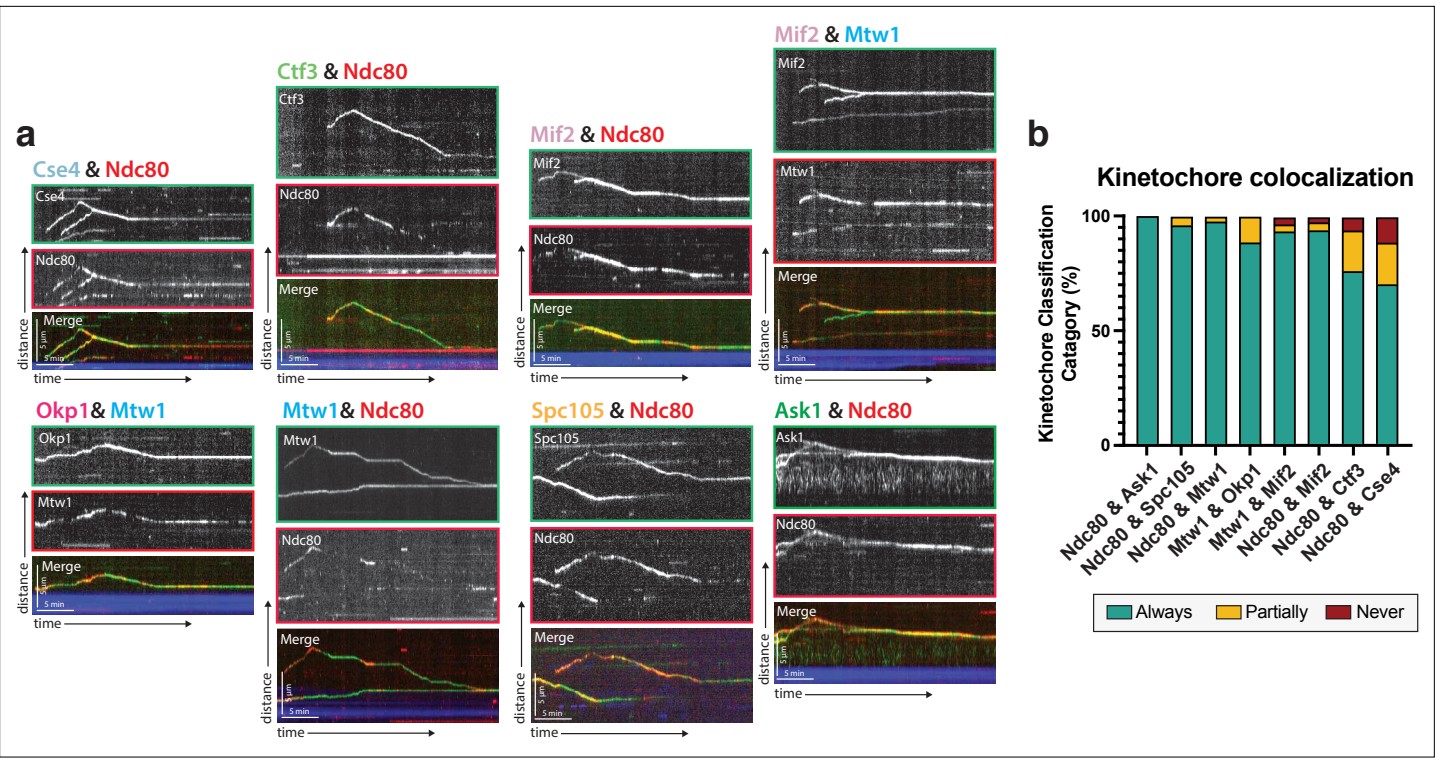

**Figure 2.** Protein composition of reconstituted kinetochores. (**a**) Different pairwise combinations of kinetochore sub-complex proteins colocalize, establishing that the kinetochore spots observed by fluorescence microscopy are multi-protein, multi-subcomplex assemblies. Representative kymographs from each strain expressing a different pair-wise combination of tagged kinetochore proteins are shown. Time is on the x-axis and distance is on the y-axis (scale bars are 5 min and 5 µm, respectively). Ndc80 was used as the common reference marker for multiple pairwise combinations. Additionally, Mtw1 was used in some pairwise combinations to show that multiple sub-complexes are present in the same kinetochore. The tracks of these colocalized proteins follow similar dynamics as in *Figure 1*, where they move processively toward the microtubule plus end. At a certain point, they reach the plus end and motility becomes minus-end directed concomitant with the onset of microtubule disassembly (note that the microtubules are not visualized due to the inability to measure 4 colors, and the assumption is that this directional switch in kinetochore protein motility corresponds to microtubule depolymerization onset shown in *Figure 1*). (**b**) Quantification of kinetochore colocalization for each pair-wise protein combination. Kinetochore tracks were grouped into three distinct pools and counted. These categories were 'always' colocalized (blue), 'partially' colocalized (yellow), or 'never' colocalized (red). For partially colocalized tracks, the mScarlet signal initially appears alone, but a GFP signal appears on it and stays colocalized for the remainder of the movie. For the different kinetochore protein pairs in order in the figure: from Ndc80 and Ask1 in to Ndc80 and Cse4, N=70, 49, 42, 52, 120, 81, 116, 97 tracks categorized, respectively, from two experimental replicates.

The online version of this article includes the following video for figure 2:

**Figure 2—video 1.** Multiple kinetochore proteins are dynamic on the microtubule together.
https://elifesciences.org/articles/78450/figures#fig2video1

lateral movement or the transition from lateral to end-on binding). But the colocalization of entire tracks over time in the kymographs (*Figure 2b*), even with one reporter signal blinking, indicated that a multi-complex of kinetochore proteins was observed on microtubules in our assay.

## Cryo-correlative light microscopy and electron tomography provides evidence that the reconstituted kinetochores are largely intact

Cryo-electron tomography (cryo-ET) is a powerful method for studying the structural organization of multi-protein complexes in their fully hydrated, native state (*Schur, 2019*). A challenge in cryo-ET is to relate specific protein localization to the complex structural information that can be obtained with this method. To overcome this challenge and to test whether the kinetochore proteins in our reconstitution assay are indeed part of higher-order kinetochore assemblies, we established a cryo-correlative light microscopy and electron tomography procedure. In brief, samples were prepared from yeast cell lysates with GFP-labeled Spc105 and mRuby2-labeled Tub1 incubated with far-red MT seeds in a similar fashion to the TIRF assay (see Materials and methods for details). Samples were vitrified and first imaged by

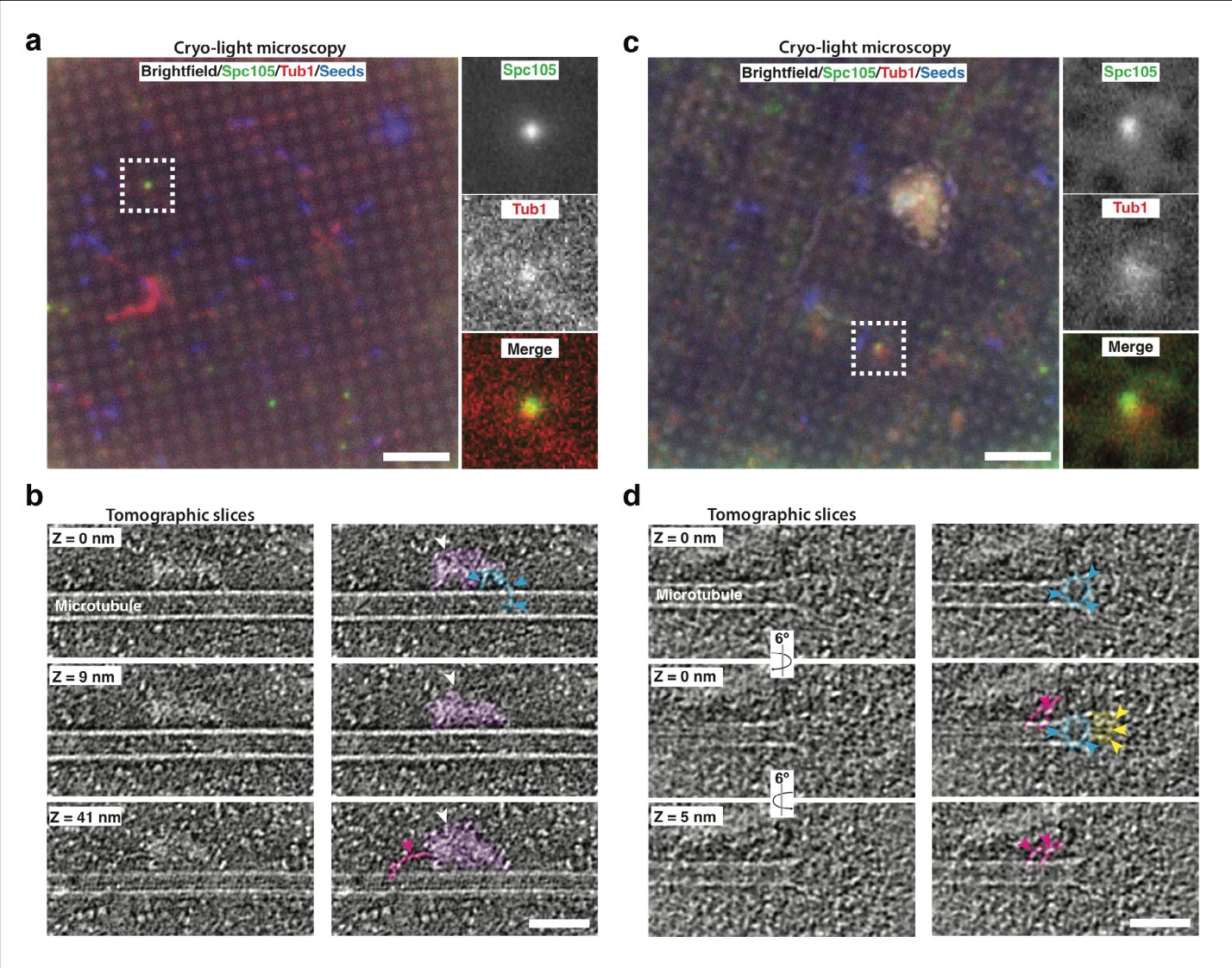

**Figure 3.** Cryo-correlative light microscopy and electron tomography shows microtubule-associated intact kinetochore complexes. (**a**, **c**) Overlay of brightfield and fluorescent channel cryo-light micrographs showing Spc105-GFP in green, mRuby-Tub1 in red and microtubule seeds in blue. The brightfield channel highlights the holey carbon film pattern. Insets show 2.7x magnified views of boxed regions. (**b**) Slices of tomogram recorded in the region of the highlighted Spc105 focus in (**a**) at Z-positions relative to the top slice. Slices show a kinetochore bound to the lateral surface of a microtubule. The largest part of the kinetochore appears as an electron-dense proteinaceous cloud similar to what has been reported for purified yeast kinetochores (white arrowheads; *Gonen et al., 2012*). A ring-like density (blue arrowheads) and a rod-like density (pink arrowheads), potentially representing Dam1 and Ndc80 sub-complexes, respectively, extend from the cloud-like density. (**d**) Slices of tomogram recorded in the region of the highlighted Spc105 focus in (**c**) at Z-positions relative to the top slice, showing a kinetochore complex bound to the end of a microtubule. Arrowheads point at a ring-like (blue) and rod-like (pink) density similar to the ones observed in (**b**) Additional rod-like extensions (yellow arrowheads) might represent the MIND complex. Right columns in (**b,d**) show the same images as in the left column, but protein densities are colored to match the arrows. Scale bars, 10 μm in (**a**, **c**), 60 nm in (**b**, **d**).

The online version of this article includes the following video for figure 3:

**Figure 3—video 1.** Cryo-electron tomogram of kinetochore bound to the lateral surface of a microtubule.

https://elifesciences.org/articles/78450/figures#fig3video1

**Figure 3—video 2.** Cryo-electron tomogram of kinetochore bound to the tip of a microtubule.
https://elifesciences.org/articles/78450/figures#fig3video2

cryo-fluorescence light microscopy to identify regions of interest for cryo-ET data collection (*Figure 3a and c*). The holey carbon film generated some fluorescent background in the GFP channel, but we could detect clear Spc105-GFP foci. These foci were then targeted for cryo-ET imaging. Large protein complexes associated with the lateral surface and the tip of individual microtubules were clearly visible in tomograms of these regions (*Figure 3—video 1*, *Figure 3—video 2*). These approximately 100 nm long complexes are reminiscent of kinetochore particles assembled from purified yeast components (*Gonen et al., 2012*; *Figure 3b and d*). Furthermore, our tomography data revealed electron densities that likely correspond to the Dam1 ring and the Ndc80 and MIND sub-complexes (*Westermann et al., 2005*; *Jenni and Harrison, 2018*; *Ng et al., 2019*; *Figure 3b and d*). In combination with the co-localization data from the TIRF assay, our tomography data reinforces the conclusion that multi-protein kinetochore particles were present on yeast microtubules in our reconstitution cell lysate assays.

## Kip3 is involved in kinetochore movement toward the microtubule plus end

After establishing that the kinetochore protein signals observed in our reconstitution assay represented intact kinetochore complexes, we began to probe the mechanism of the plus-end directed kinetochore movement. We postulated that the movement was driven by a kinesin motor protein. In budding yeast, there are only 5 kinesins, Kip2, Kip3, Cin8, Kip1, and Kar3 (*Cottingham et al., 1999*). To test the possibility that one of these motors was involved in kinetochore movement along microtubules in our assay, we systematically deleted each kinesin and asked whether kinetochore motility was impaired.

Upon deleting the genes encoding each of these motors, we found that only the highly processive plus-end directed kinesin-8, Kip3, was necessary for plus-end directed motility of the kinetochores (*Figure 4—video 1*). Specifically, 77% of laterally bound kinetochores moved in Kip3 +lysates compared to only 28% in *kip3Δ* lysates. Moreover, those laterally bound kinetochores that did move in *kip3Δ* lysates spent significantly less time moving toward the microtubule plus end compared to wild-type, and their run lengths were about half as far as those in Kip3 +lysates (0.46 μm vs 0.73 μm; *Figure 4b and c*). Interestingly, although less frequent and covering shorter distances, kinetochore movements in *kip3Δ* lysates had the same velocity as the more frequent and longer movements observed in Kip3 +lysates, moving at 0.47 μm/min (*Figure 4d*; see Discussion).

Because the kinetochore movements in the *kip3Δ* lysate were relatively infrequent and covered short distances, only 5.7% of laterally bound kinetochores reached the plus end, compared to 28.1% in Kip3 +lysates. As in the Kip3 +lysates, when the kinetochore in the *kip3Δ* lysates was bound to the tip of the microtubule, depolymerization with the kinetochore tracking the microtubule plus end did occur. However, in this *kip3Δ* case, the microtubule plus ends with tip-bound kinetochores spent a significantly lower percentage of the time depolymerizing compared to what we observed in Kip3 +lysates (*Figure 4b*). This observation is consistent with a role for Kip3 in kinetochore-induced microtubule depolymerization (*Gupta et al., 2006*).

Although Kip3 is required for kinetochore motility, whether kinetochores that were assembled and bound to the lateral surface of microtubules in the absence of the kinesin could be induced to move processively by retroactive addition of Kip3 was not clear. It was possible that an important binding surface for the kinesin or its cargo would only be accessible when the kinetochore was partially assembled or unattached to microtubules. Therefore we tested whether providing a source of wild-type Kip3 to stationary kinetochores on microtubules assembled in *kip3Δ* lysates would enable those kinetochores to start to move. After observing immobile kinetochores on microtubules assembled in *kip3Δ* lysates for 10 min, lysate from a Kip3 +strain was added to the chamber. Immediately, these same stationary kinetochores began to move processively toward microtubule plus ends, establishing that Kip3 acts on previously microtubule-bound kinetochores (*Figure 4e*). Specifically, 53.7% of the 54 immobile, laterally bound kinetochores in the *kip3Δ* lysates became motile after Kip3 +lysate was added. Therefore, kinetochores assembled in the absence of Kip3 were able to bind to microtubules, but required the subsequent addition of Kip3 and/or its other binding partners to translocate along the lateral surface of microtubules.

## Processive Kip3 transiently encounters slower moving kinetochores

We next set out to investigate the molecular mechanism by which Kip3 facilitates kinetochore plus end movement. To start, we asked whether we could observe Kip3 stably associated with moving

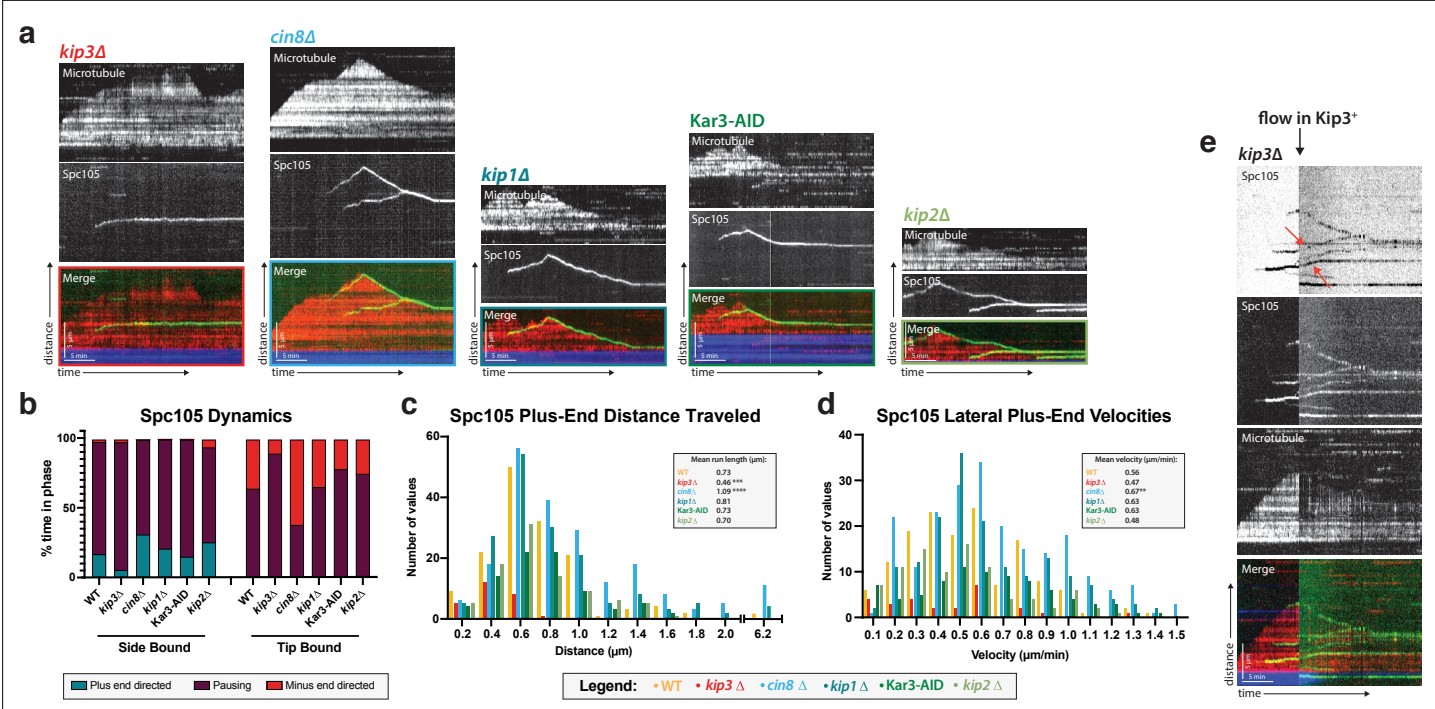

**Figure 4.** Kip3, a kinesin-8 motor, is required for plus-end directed movement of the kinetochore on the microtubule lattice. (**a**) Genes encoding each of the five yeast kinesins were individually deleted (or the proteins degraded using a degron tag) and kinetochore dynamics on microtubules were assayed. Only Kip3, a highly processive kinesin-8, was required for kinetochore movement toward the microtubule plus end. Representative kymographs are shown with time on the x-axis and distance on the y-axis (scale bars are 5 min and 5 μm, respectively). The kinetochore (Spc105) is shown in green and the microtubule is shown in red. The microtubule seed (at the minus-end of the growing red yeast microtubule) is shown in blue. (**b**) Quantification of dynamics over time shows that kinetochores in the *kip3Δ* strain spend the majority of the time paused, moving toward the plus end only 5% of the time compared to 30% for wild-type. (**c**) Run lengths and the number of runs the kinetochore made on the lattice toward the plus end were decreased in the *kip3Δ* strain. Run lengths were about half of wild-type, 0.46 μm vs 0.73 μm. The number of runs was also decreased to about 25% of the wild-type number of runs. Statistical analysis was done by a Kruskal-Wallis test where *** is p=0.0001 and **** is p<0.0001. (**d**) Laterally bound kinetochores velocities are unchanged when moving in the plus end direction in a *kip3Δ* lysate. Statistical analysis was done using a Kruskal-Wallis test where ** is p=0.0047. (**b–d**) Quantification is from four replicate trials of two biological replicates. For each strain from WT to *kip2Δ* lysate in the order listed, N=128, 70, 129, 144, 103, 154 Spc105 proteins tracked. (**e**) Adding back a source of Kip3 to a lysate from a *kip3Δ* strain restores plus end-directed kinetochore movement. In lysates lacking Kip3, Spc105 (green) was immobile on the microtubule (red) surface. When Kip3 +lysate was added, the same, previously immobile, kinetochores started to translocate toward the microtubule plus ends (marked with red arrows).

The online version of this article includes the following video and source data for figure 4:

**Source data 1.** Source data supplied.

**Figure 4—video 1.** Kinetochores do not move when the Kip3 gene is deleted.

https://elifesciences.org/articles/78450/figures#fig4video1

---

kinetochores. To achieve this goal, we carried out dual-color imaging of the kinetochore (Spc105) and each of the motors assayed in the motor deletion screen described above. We observed Cin8 and Kip1 coming on and off the microtubule lattice stochastically at a high frequency, but they were not motile in our assay (*Figure 5—figure supplement 1a*). Kar3 was rarely observed on the microtubule (*Figure 5—figure supplement 1a*). For these three motors, anti-parallel or parallel bundled microtubules might be necessary to observe their previously reported activities in lysate (*Su et al., 2013*; *Hepperla et al., 2014*).

Of all the motors assayed, only Kip3 and Kip2 moved processively on the microtubule lattice (*Figure 5—figure supplement 1a*). However, kinetochore motility was only disrupted in *kip3Δ* lysates, not in *kip2Δ* lysates (*Figure 4a–d*). Therefore, processive motor movement towards the plus end alone is not sufficient to facilitate kinetochore movement along a microtubule. Kinetochore plus-end directed movement is dependent on Kip3 activity specifically. However, neither motor showed persistent colocalization with the kinetochores. The dual color tagging of Kip3 and Spc105 showed

that there were brief moments of kinetochore movement where Kip3 signal was observed running over or through the kinetochore signal (*Figure 5—video 1*). Kip3 also moved much faster than either the speed of the kinetochore or the polymerization of the microtubule. However, Kip3 density on the microtubule was also very high and the bright signal might have obscured any Kip3 associated stably with kinetochores (see Discussion).

## Kip3's tail domain is not needed for kinetochore motility, but Kip3's ability to attenuate microtubule dynamics is needed for kinetochores to reach the microtubule plus end

Comparison of in vitro and in vivo phenotypes of different Kip3 mutants might provide insights into the role of Kip3 in establishment of end-on attachments. In cells, knocking out the gene encoding Kip3 results in metaphase kinetochores that stay along the axis of the spindle, but are no longer focused into two discrete foci (*Figure 5a–b*; *Tytell and Sorger, 2006*; *Wargacki et al., 2010*; *Su et al., 2011*). This declustered kinetochore phenotype has been proposed to be a result of improper spindle assembly and/or defective kinetochore-microtubule attachment (*Shimogawa et al., 2010*). Of interest to us was a Kip3 mutant, Kip3ΔT-LZ, that shows a declustered kinetochore phenotype in vivo (*Figure 5a–b*; *Su et al., 2011*). In Kip3ΔT-LZ, the putative cargo-binding tail was truncated and replaced with a leucine zipper domain from Gcn4 transcription factor to restore dimerization of the motor heads (*Su et al., 2011*). Because the declustered chromosome phenotype in vivo is believed to be due to defects in kinetochore attachment, we hypothesized that, in our in vitro assay, we would observe laterally attached kinetochores in *kip3ΔT-LZ* lysates that do not become properly end-bound, similar to what we observed in *kip3Δ* lysates. Strikingly, in *kip3ΔT-LZ* lysates, kinetochores still moved processively toward the plus end of microtubules (*Figure 5—video 2*). In fact, deleting the tail domain increased the velocity and the average run length of the kinetochore on the lattice of the microtubule (*Figure 5d*). In Kip3 +versus *kip3ΔT-LZ* lysate, the laterally bound plus end velocity increased from 0.56 µm/min to 0.76 µm/min and the average run length increased from 0.56 µm/min to 1.4 µm/min (*Figure 5d*).

As previously reported, the Kip3ΔT-LZ mutant protein did not accumulate on the plus ends of microtubules and it was unable to destabilize their plus ends and cause depolymerization (*Su et al., 2011*; *Figure 5—figure supplement 1b*). Similarly, in a *kip3Δ* lysate, there was also altered microtubule dynamics, specifically a diminished catastrophe frequency (*Bergman et al., 2018*; *Figure 5—figure supplement 1b*). Because of the reduced microtubule catastrophe frequency, we observed that in *kip3ΔT-LZ* lysates the kinetochore rarely reached the plus ends of polymerizing microtubules, despite its faster velocity. Moreover, compared to the Kip3 +lysates wherein 28.1% of kinetochores completed travel to the microtubule plus end, only 12.4% of kinetochores made it to microtubule plus ends in *kip3ΔT-LZ* lysates. The decreased kinetochore movement to the microtubule plus ends was even more dramatic in *kip3Δ* lysates, where only 5.7% of kinetochores reached the plus end (*Figure 5c*). These results indicate that the Kip3 tail domain is not required for kinetochore movement along microtubules, but it is required for kinetochores to reach the plus end. We conclude that Kip3's dual functions are: (1) attenuating microtubule growth and (2) enabling movement of kinetochores processively on the lattice to allow end-on attachments to be made (see Discussion).

## Discussion

Here we present a novel assay to follow kinetochore movement on microtubules in a cell-free yeast system. We demonstrate that this assay allows individual kinetochores to be visualized while they are interacting with single microtubules. We observed kinetochore binding to microtubules, movement toward the microtubule plus end along the lattice, transition from side- to end-on attachment, apparently inducing depolymerization, and tracking of the depolymerizing end. Development of this kinetochore assay was achieved by adopting a previously established yeast extract system that allows analysis of dynamics of single yeast microtubules assembled in the presence of the full complement of yeast proteins at defined cell cycle stages (*Bergman et al., 2018*). This assay involves native yeast proteins and allows us to combine the control over conditions provided by complementary powers of biochemistry and genetics to establish functions. One limitation of this system that we hope to address in the future is that molecules that are normally compartmentalized to either the nucleus or

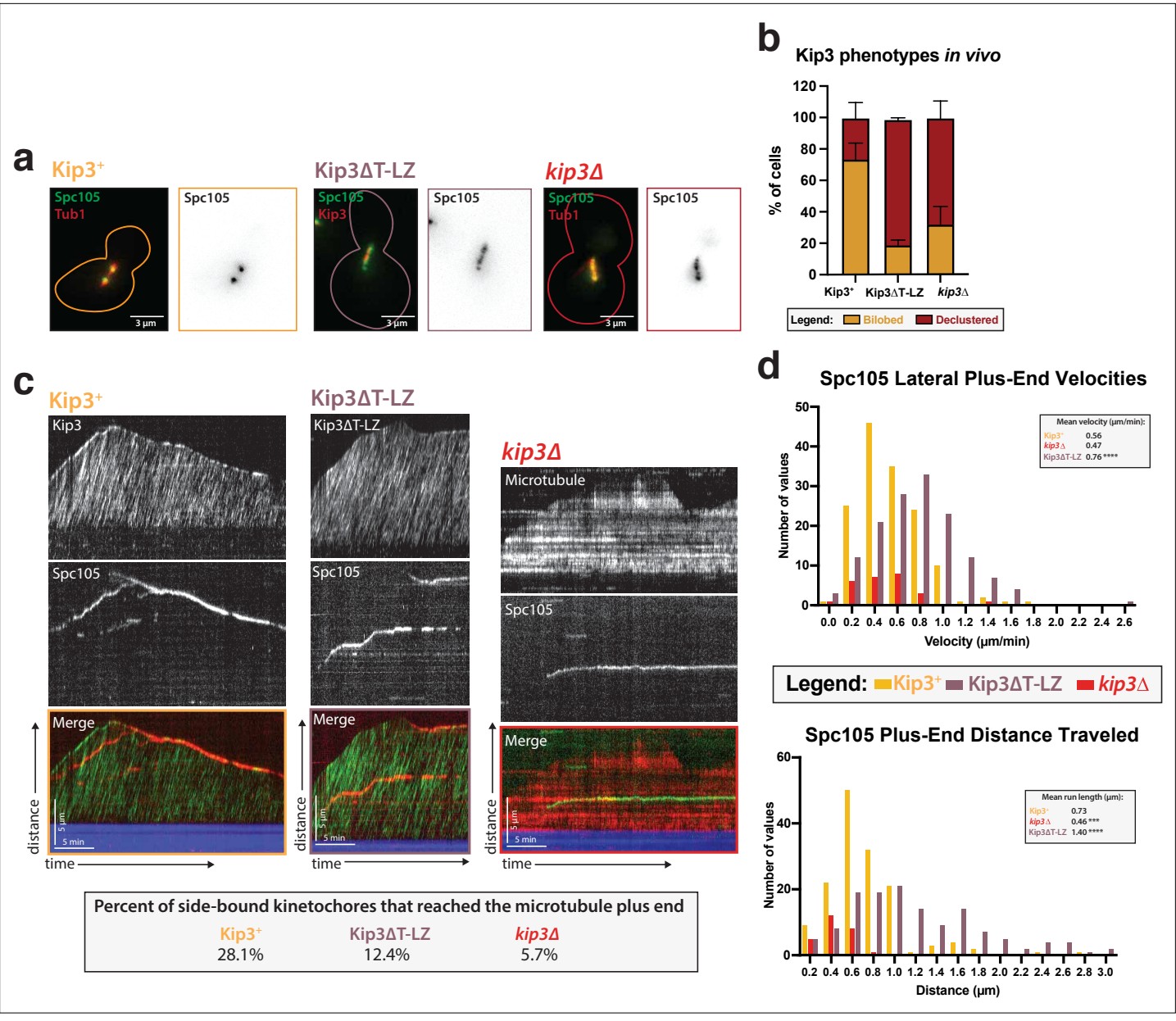

**Figure 5.** Kip3's motor activity, but not its depolymerase tail domain, is necessary for processive plus end-directed kinetochore movement. (**a**) Metaphase cells with either wild-type Kip3, *kip3Δ*, or *kip3ΔT-LZ* alleles imaged in a *cdc23-1* strain expressing Spc105-GFP (green), mRuby-Tub1 or Kip3 (red). Note that the cells were not shifted to non-permissive temperature and were not arrested for imaging. Images are a maximum Z-projection of a 5 μm stack of 0.2 μm slices. (**b**) Quantification of the declustered kinetochore phenotype is shown as a bar graph for spindles that were 2–3 μm in length. Cells were classified as 'bi-lobed' by a line scan showing two distinct peaks. The 'declustered' phenotype included all line scans that did not show two peaks (line scans not shown). Two replicates were performed, each with n=50 cells. Error bars are standard error of the mean. (**c**) Representative kymographs of Spc105 motility in wild-type, *kip3ΔT-LZ*, and *kip3Δ* lysates. Dual color tagging of wild-type Kip3 and Spc105 showed no clear colocalization between the kinetochore (red) and Kip3 (green). Kip3 is highly processive, moving much faster than the kinetochore. A Kip3 mutant containing a deletion in the tail region (Kip3ΔT-LZ, green) with the kinetochore (red) was still processive along the microtubule. The Kip3ΔT-LZ mutant showed robust processivity but it did not accumulate on microtubule plus ends and depolymerization events were fewer than in Kip3 +lysates, as shown in previously published literature. Lastly, a representative kymograph for *kip3Δ* lysate shows immobile Spc105 (green) on microtubules (red). (**d**) Histograms of Spc105 velocities and average run length when laterally bound and moving toward the microtubule plus ends. The number of values is shown on the y-axis and velocity/distance traveled on the x-axis, binned every 0.2 units. The mean velocity of wild-type Kip3 protein and Kip3ΔT-LZ protein is 0.55 μm/min and 0.76 μm/min, respectively. The average run length also increased from 0.73 μm to 1.4 μm. The mean Spc105 velocity in *kip3Δ* lysates was unchanged compared to wild-type, but the number of runs was reduced to 25% of wild-type. The average length decrease in *kip3Δ* was reduced from 0.73 μm to 0.45 μm. The mean is from two replicate trials of one biological replicate. Statistical analysis was done using a Kruskal-Wallis test where *** p=0.0006 and **** p<0.0001. For lysates from each strain, as listed from WT to Kip3ΔT-LZ, N=128, 70, 81 kinetochores tracked.

*Figure 5 continued on next page*

*Figure 5 continued*

The online version of this article includes the following video, source data, and figure supplement(s) for figure 5:

**Source data 1.** Source data supplied.

**Figure supplement 1.** Kinetochores do no colocalize with the other four kinesins and Kip3's effect on microtubule dynamics.

**Figure 5—video 1.** Kinetochore and Kip3 dynamics Spc105 (red) and Kip3 (green) move on the lattice of the microtubule towards the plus-end.

https://elifesciences.org/articles/78450/figures#fig5video1

**Figure 5—video 2.** Kinetochore and Kip3ΔT-LZ dynamics Spc105 (red) and Kip3ΔT-LZ (green) move on the lattice of the microtubule towards the plus-end.

https://elifesciences.org/articles/78450/figures#fig5video2

the cytoplasm are mixed. We propose a model wherein kinetochores move to the microtubule plus ends powered by the budding yeast kinesin-8, Kip3. When kinetochores reach the microtubule plus ends, their attachments are converted to end-on attachments, at which point microtubule disassembly ensues (*Figure 6*).

Using both TIRF microscopy and cryo-electron tomography, we verified successful reconstitution of intact, microtubule-bound kinetochores. We observed nine different kinetochore proteins, spanning five kinetochore sub-complexes, and found that they all associate with and move along microtubules with similar dynamics (*Figure 1a–e*). The common behavior of these kinetochore proteins, combined with their colocalization with each other (*Figure 2*), indicated that kinetochore particles that contain multiple subunits from different subcomplexes were observed in the assay. Since our assay does not involve biochemical fractionation steps, and since the lysates retain the relative physiological concentration and ratio of soluble proteins, we were able to preserve the natural kinetochore composition (*Biggins et al., 2013*). Most importantly, by using cryo-correlative light-microscopy and electron tomography, we were able to directly visualize kinetochores bound to both the sides and ends of microtubules and found that their structures closely resemble negatively stained yeast kinetochores observed in earlier studies (*Gonen et al., 2012*). In both side- and end-bound arrangements, these kinetochore complexes included rod-like and ring-like structures that most likely are the Ndc80 complex and the Dam1 complex, respectively (*Figure 3a–d*, *Gonen et al., 2012*; *Westermann et al., 2005*; *Jenni and Harrison, 2018*; *Ng et al., 2019*). Taken together, these observations establish that we were able to recapitulate dynamic kinetochore-microtubule binding in an in vitro reconstitution assay.

The ability to visualize individual kinetochores associated with single microtubules made possible quantitative analysis of their dynamics along microtubules, which mimicked kinetochore behavior observed in spindles in vivo (*Maiato et al., 2017*). Kinetochores that bind to the side of the microtubule lattice move processively towards the plus end. Upon reaching the microtubule end, kinetochores convert their attachment to an end-bound conformation. During their conversion from lateral- to end-on attachment, and while bound to the microtubule end, kinetochores remain stably attached to the microtubule. Once bound to the microtubule ends, kinetochores track the tip of the depolymerizing microtubule. This transition to stable end-on attachments appeared to block microtubule polymerization and to induce depolymerization at a rate slower than the catastrophe rate for microtubules lacking kinetochores (*Bergman et al., 2018*, *Figure 1a–e*). While tension stabilizes kinetochore-microtubule attachments on dynamic microtubules (*Akiyoshi et al., 2010*), in our presumably tension-less system, there was no observable kinetochore detachment from microtubule plus ends. Differences in kinetochore composition, cell-cycle-specific post-translational modifications, species-specific tubulin, buffer composition, or the presence of unidentified helper proteins could account for the differences. Notably, end-bound kinetochores do not normally detach from microtubules in vivo after attachment has been established during S phase (*Tanaka et al., 2005b*). Additionally, in a previous in vitro study, chromosomes remained stably associated with depolymerizing microtubule ends (*Koshland et al., 1988*). In the future, adapting our assay to include tension across the kinetochore promises to provide insights into how force impacts attachment and motility parameters. In summary, we were able to reconstitute intact kinetochores that associated with microtubule lateral surfaces and then underwent conversion to end-on attachment, which was followed by Anaphase A-like kinetochore-microtubule dynamics, with the kinetochores staying associated with depolymerizing microtubule plus ends.

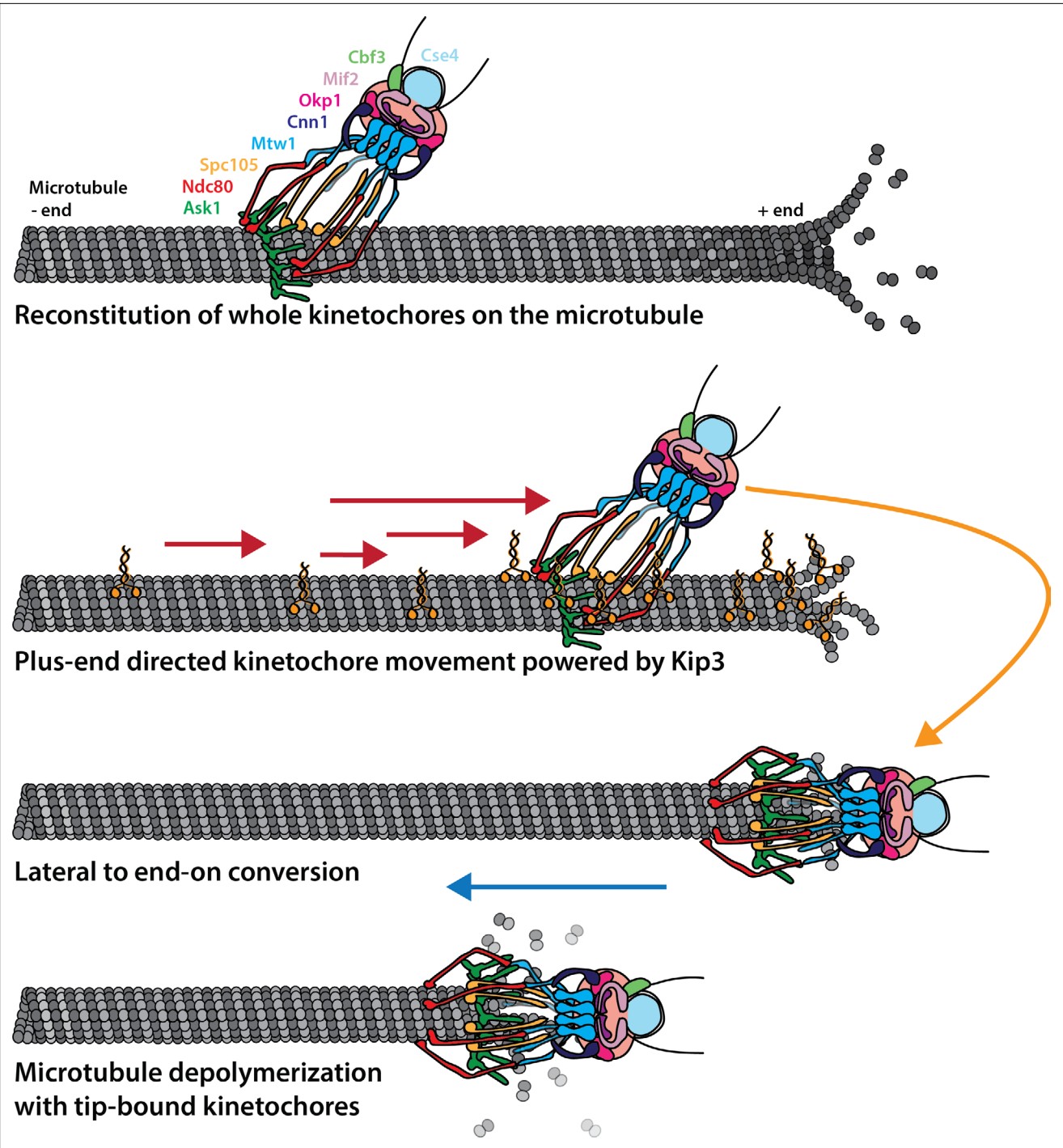

**Figure 6.** Cartoon model of reconstituted kinetochore dynamics on the microtubule Yeast kinetochores reconstituted binding with individual yeast microtubules in a yeast protein extract. These laterally-bound kinetochores travel in a directional manner to the plus end of the microtubule, powered by the kinesin-8, Kip3. Upon reaching the end of the microtubule, the kinetochore transitions to end-on attachment. Establishment of this end-on attachment coincides with onset of microtubule depolymerization and the end-bound kinetochore preventing the microtubule from elongating again.

Strikingly, reconstituted intact kinetochores associated with the lateral surface of dynamic microtubules stably, and they moved exclusively toward the plus end, and in a processive and kinesin-8-dependent manner. This behavior contrasts with the in vivo observation of chromosomes binding to the lateral surface of microtubules and moving exclusively towards the minus-end in a Kar3-dependent manner (*Tanaka et al., 2005b*). We speculate that these two distinct and opposite-directed

mechanisms both exist in yeast to fulfill different functions. In mammalian cells, dynein has been reported to transport chromosomes captured by astral microtubules toward the centrosome, while CENP-E is implicated in moving laterally bound chromosomes in the plus end direction toward the spindle equator to promote amphitelic sister kinetochore attachment (*Maiato et al., 2017*). We propose that in yeast, minus-end-directed Kar3 may transport chromosomes toward the spindle pole, while Kip3 moves chromosomes toward the equator. These two mechanisms may increase resilience to different stresses. Further studies are required to fully explore how these two mechanisms are employed in normal and stressed conditions.

In our assay, kinetochores reached the plus end via a combination of directed movement and microtubule catastrophe. A role for Kip3, the budding yeast kinesin-8, in the translocation of kinetochores along microtubules is consistent with Kip3's reported in vivo colocalization with kinetochores and with its Ndc10-dependent association with centromeric DNA in ChIP assays (*Tytell and Sorger, 2006*). While our experiments did not show strong colocalization of Kip3 and kinetochores, it is possible that weak transient interactions are what facilitates kinetochore mobility via Kip3 (*Figure 5c*), or that low stoichiometry of Kip3 stably associated with kinetochores made detection impossible against a high background (*Figure 5c*) of free Kip3 molecules on the microtubules. Additionally, quantification of kinetochore velocity on the microtubule lattice showed a wide distribution of rates punctuated by frequent pausing (*Figure 1d–e*). Both of these observations lend themselves to a model in which Kip3 affects kinetochore mobility in a stochastic way, perhaps by binding transiently. By this model, Kip3 sometimes moves the kinetochore in longer, faster runs, while other times, it displaces the kinetochore a short distance. Significantly, another processive microtubule-plus-end directed motor protein, Kip2 (*Figure 5—figure supplement 1b*), was insufficient to move kinetochores in *kip3Δ* lysates, which diminishes the possibility that non-specific collisions with the motor proteins are driving kinetochore translocation. The plus-end directed kinetochore motility we observed is specifically dependent on a processive Kip3 motor and was not dependent on this kinesin's tail domain. In the absence of the ability to consistently observe direct Kip3 stable association with motile kinetochores, we cannot rule out other mechanisms to explain the Kip3-dependent kinetochore motility that we discovered. Further investigation is required to determine the exact molecular mechanism by which Kip3 mediates plus end-directed kinetochore transport on the microtubule lattice. It is possible, for example, that Kip3 transports an unidentified factor to kinetochores, and that this hypothetical factor mediates kinetochore movement.

## Ideas and speculation

We propose that Kip3 has a dual function in facilitating end-on attachment: (1) it induces microtubule catastrophe to allow kinetochores to catch up with elongating microtubule plus ends and (2) it moves kinetochores to the microtubule plus ends. Kinetochores did not move as frequently, directionally, or as far along microtubules in the *kip3Δ* lysate, but they did move in a highly processive manner in *kip3ΔT-LZ* lysates (*Figure 5c–d*). However, in lysates made from both Kip3 mutants (*kip3Δ* or *kip3ΔT-LZ*), we observed kinetochores that are laterally attached to microtubules, but these kinetochores rarely became end-bound. This observation can be explained if fewer kinetochores make it to the end of the microtubule. Specifically, in *kip3Δ* lysates, the inability to establish end-on attachments is likely largely due to a combination of impaired kinetochore motility and due to the microtubule dynamics being shifted toward more elongation with fewer catastrophes, making it less likely that kinetochores will reach microtubule plus ends (*Figure 4a–d*, *Figure 5—figure supplement 1b*). In contrast, in *kip3ΔT-LZ* lysates the kinetochores are highly motile, but they establish end-on attachments less frequently than in wild-type lysates. This result can be explained by the observations that the catastrophe frequency is severely diminished compared to wild-type microtubule dynamics, and that the microtubules mostly elongate faster than the rate of kinetochore translocation, so the kinetochores are less likely to encounter plus ends (*Figure 5c–d*, *Figure 5—figure supplement 1b*).

It is useful to consider these in vitro results in the context of observations of kinetochore behavior in the corresponding mutant cells. In wild-type cells, sister chromatids are clustered in two foci within the metaphase spindle. In contrast, in *kip3Δ* cells metaphase kinetochores are declustered and aligned linearly along the spindle microtubules. This declustered kinetochore phenotype is also seen in *kip3ΔT-LZ* cells (*Tytell and Sorger, 2006*; *Wargacki et al., 2010*; *Su et al., 2011*; *Figure 5b*). Based on the above in vitro and in vivo observations, we hypothesize that laterally bound kinetochores are

an intermediate on the pathway toward biorientation, and that Kip3 is required for moving laterally attached kinetochores to the plus end of the microtubule during prometaphase. Specifically, we speculate that the previously reported declustered kinetochore phenotype is the result of an increased number of laterally bound kinetochores stalled all along spindle microtubules.

# Materials and methods

## Key resources table

| Reagent type (species) or resource | Designation | Source or reference | Identifiers | Additional information |
|---|---|---|---|---|
| Strain, strain background (*Saccharomyces cerevisiae*) | cdc23-1, MIF2-GFP, mRuby2-TUB1 | This paper | DDY 5806 | MATa, cdc23-1, MIF2-GFP::KanMX, TUB1::PHIS3-mRuby2-TUB1::HPH, his3-Δ200, leu2-3,112, ura3-52 |
| Strain, strain background (*Saccharomyces cerevisiae*) | cdc23-1, MIF2-GFP, mRuby2-TUB1 | This paper | DDY 5807 | MATalpha, cdc23-1, MIF2-GFP::KanMX, TUB1::PHIS3-mRuby2-TUB1::HPH, his3-Δ200, leu2-3,112, lys2-801(am), ura3-52 |
| Strain, strain background (*Saccharomyces cerevisiae*) | cdc23-1, MTW1-GFP, mRuby2-TUB1 | This paper | DDY 5808 | MATa, cdc23-1, MTW1-GFP::KanMX, TUB1::PHIS3-mRuby2-TUB1::HPH, his3-Δ200, leu2-3,112, lys2-801(am), ura3-52 |
| Strain, strain background (*Saccharomyces cerevisiae*) | cdc23-1, MTW1-GFP, mRuby2-TUB1 | This paper | DDY 5809 | MATalpha, cdc23-1, MTW1-GFP::KanMX, TUB1::PHIS3-mRuby2-TUB1::HPH, his3-Δ200, leu2-3,112, lys2-801(am), ura3-52 |
| Strain, strain background (*Saccharomyces cerevisiae*) | cdc23-1, CSE4-GFP, mRuby2-TUB1 | This paper | DDY 5810 | MATa, cdc23-1, CSE4-3x flexible linker-yoEGFP::KanMX, mRuby2-TUB1::URA3, his3-Δ200, leu2-3,112, lys2-801(am) |
| Strain, strain background (*Saccharomyces cerevisiae*) | cdc23-1, CSE4-GFP, mRuby2-TUB1 | This paper | DDY 5811 | MATalpha, cdc23-1, CSE4-3x flexible linker-yoEGFP::KanMX, mRuby2-TUB1::URA3, his3-Δ200, leu2-3,112, lys2-801(am) |
| Strain, strain background (*Saccharomyces cerevisiae*) | cdc23-1, OKP1-GFP, mRuby2-TUB1 | This paper | DDY 5812 | MATa, cdc23-1, OKP1-yoEGFP::KanMX, URA3::mRuby2-TUB1, his3-Δ200, leu2-3,112, lys2-801(am) |
| Strain, strain background (*Saccharomyces cerevisiae*) | cdc23-1, OKP1-GFP, mRuby2-TUB1 | This paper | DDY 5813 | MATalpha, cdc23-1, OKP1-yoEGFP::KanMX, URA3::mRuby2-TUB1, his3-Δ200, leu2-3,112, lys2-801(am) |
| Strain, strain background (*Saccharomyces cerevisiae*) | cdc23-1, CTF3-GFP, mRuby2-TUB1 | This paper | DDY 5814 | MATa, cdc23-1, CTF3-yoEGFP::KanMX, URA3::mRuby2-TUB1, his3-Δ200, leu2-3,112, lys2-801(am) |
| Strain, strain background (*Saccharomyces cerevisiae*) | cdc23-1, CTF3-GFP, mRuby2-TUB1 | This paper | DDY 5815 | MATalpha, cdc23-1, CTF3-yoEGFP::KanMX, URA3::mRuby2-TUB1, his3-Δ200, leu2-3,112, lys2-801(am) |
| Strain, strain background (*Saccharomyces cerevisiae*) | cdc7-1, SPC105-GFP, mRuby2-TUB1 | This paper | DDY 5816 | MATa, cdc7-1, SPC105-GFP::KanMX, mRuby2-TUB1::HPH, his3-Δ200, leu2-3,112, lys2-801(am), ura3-52 |
| Strain, strain background (*Saccharomyces cerevisiae*) | cdc15-2, SPC105-GFP, mRuby2-TUB1 | This paper | DDY 5817 | MATa, cdc15-2, SPC105-GFP::KanMX, mRuby2-TUB1::HPH, his3-Δ200, leu2-3,112, ura3-52 |
| Strain, strain background (*Saccharomyces cerevisiae*) | cdc23-1, SPC105-GFP, mRuby2-TUB1 | This paper | DDY 5818 | MATa, cdc23-1, SPC105-3x flexible linker-yoEGFP::KanMX, TUB1::PHIS3-mRuby2-TUB1::HPH, his3-Δ200, leu2-3,112, lys2-801(am), ura3-52 |
| Strain, strain background (*Saccharomyces cerevisiae*) | cdc23-1, SPC105-GFP, mRuby2-TUB1 | This paper | DDY 5819 | MATalpha, cdc23-1, SPC105-3x flexible linker-yoEGFP::KanMX, TUB1::PHIS3-mRuby2-TUB1::HPH, his3-Δ200, leu2-3,112, lys2-801(am), ura3-52 |
| Strain, strain background (*Saccharomyces cerevisiae*) | cdc23-1, SPC105-GFP, kip3Δ, mRuby2-TUB1 | This paper | DDY 5820 | MATa, cdc23-1, SPC105-3x flexible linker-yoEGFP::KanMX, kip3Δ::HIS3MX6, TUB1::PHIS3-mRuby2-TUB1::HPH, his3-Δ200, leu2-3,112, lys2-801(am), ura3-52 |
| Strain, strain background (*Saccharomyces cerevisiae*) | cdc23-1, SPC105-GFP, kip3Δ, mRuby2-TUB1 | This paper | DDY 5821 | MATalpha, cdc23-1, SPC105-3x flexible linker-yoEGFP::KanMX, kip3Δ::HIS3MX6, TUB1::PHIS3-mRuby2-TUB1::HPH, his3-Δ200, leu2-3,112, lys2-801(am), ura3-52 |
| Strain, strain background (*Saccharomyces cerevisiae*) | cdc23-1, SPC105-GFP, cin8Δ, mRuby2-TUB1 | This paper | DDY 5822 | MATa, cdc23-1, SPC105-3x flexible linker-yoEGFP::KanMX, cin8Δ::HIS3MX6, TUB1::PHIS3-mRuby2-TUB1::HPH, his3-Δ200, leu2-3,112, lys2-801(am), ura3-52 |

*Continued on next page*

*Continued*

| Reagent type (species) or resource | Designation | Source or reference | Identifiers | Additional information |
|---|---|---|---|---|
| Strain, strain background (*Saccharomyces cerevisiae*) | cdc23-1, SPC105-GFP, cin8Δ, mRuby2-TUB1 | This paper | DDY 5823 | MATalpha, cdc23-1, SPC105-3x flexible linker-yoEGFP::KanMX, cin8Δ::HIS3MX6, TUB1::PHIS3-mRuby2-TUB1::HPH, his3-Δ200, leu2-3,112, lys2-801(am), ura3-52 |
| Strain, strain background (*Saccharomyces cerevisiae*) | cdc23-1, NDC80-mScarlet-I, MTW1-GFP, HaloTag-TUB1 | This paper | DDY 5824 | MATa, cdc23-1, NDC80-mScarlet-I::LEU2, MTW1-GFP::KanMX, ura3-52::PHIS3-HaloTag-flexible linker-TUB1::URA3, his3-Δ200, leu2-3,112, lys2-801(am) |
| Strain, strain background (*Saccharomyces cerevisiae*) | cdc23-1, NDC80-mScarlet-I, SPC105-GFP, HaloTag-TUB1 | This paper | DDY 5825 | MATa, cdc23-1, NDC80-mScarlet-I::LEU2, SPC105-3x flexible linker-yoEGFP::KanMX, ura3-52::PHIS3-HaloTag-flexible linker-TUB1::URA3, his3-Δ200, leu2-3,112, lys2-801(am) |
| Strain, strain background (*Saccharomyces cerevisiae*) | cdc23-1, NDC80-mScarlet-I, ASK1-GFP, HaloTag-TUB1 | This paper | DDY 5826 | MATa, cdc23-1, NDC80-mScarlet-I::LEU2, ASK1-yoEGFP::KanMX, ura3-52::PHIS3-HaloTag-flexible linker-TUB1::URA3, his3-Δ200, leu2-3,112, lys2-801(am) |
| Strain, strain background (*Saccharomyces cerevisiae*) | cdc23-1, NDC80-mScarlet-I, MIF2-GFP, HaloTag-TUB1 | This paper | DDY 5827 | MATa, cdc23-1, NDC80-mScarlet-I::LEU2, MIF2-GFP::KanMX, ura3-52::PHIS3-HaloTag-flexible linker-TUB1::URA3, his3-Δ200, leu2-3,112 |
| Strain, strain background (*Saccharomyces cerevisiae*) | cdc23-1, MTW1-mScarlet-I, OKP1-GFP, HaloTag-TUB1 | This paper | DDY 5828 | MATalpha, cdc23-1, MTW1-mScarlet-I::LEU2, Okp1-yoEGFP::KanMX, ura3::PHis3-HaloTag-TUB1::URA3, his3-Δ200, leu2-3,112, lys2-801(am) |
| Strain, strain background (*Saccharomyces cerevisiae*) | cdc23-1, NDC80-mScarlet-I, CSE4-GFP, HaloTag-TUB1 | This paper | DDY 5829 | MATa, cdc23-1, NDC80-mScarlet-I::LEU2, CSE4-3x flexible linker-yoEGFP::KanMX, ura3-52::PHIS3-HaloTag-flexible linker-TUB1::URA3, his3-Δ200, leu2-3,112, lys2-801(am) |
| Strain, strain background (*Saccharomyces cerevisiae*) | cdc23-1, NDC80-mScarlet-I, CTF3-GFP, HaloTag-TUB1 | This paper | DDY 5830 | MATa, cdc23-1, NDC80-mScarlet-I::LEU2, CTF3-yoEGFP::KanMX, ura3-52::PHIS3-HaloTag-flexible linker-TUB1::URA3, his3-Δ200, leu2-3,112, lys2-801(am) |
| Strain, strain background (*Saccharomyces cerevisiae*) | cdc23-1, SPC105-mScarlet-I, KIP3-GFP, HaloTag-TUB1 | This paper | DDY 5831 | MATa, cdc23-1, SPC105-mScarlet-I::LEU2, KIP3-yoEGFP::KanMX, ura3-52::PHIS3-HaloTag-TUB1::URA3, his3-Δ200, leu2-3,112, lys2-801(am) |
| Strain, strain background (*Saccharomyces cerevisiae*) | cdc23-1, SPC105-mScarlet-I, CIN8-GFP, HaloTag-TUB1 | This paper | DDY 5832 | MATalpha, cdc23-1, SPC105-mScarlet-I::LEU2, CIN8-GFP::KanMX, ura3-52::PHIS3-HaloTag-TUB1::URA3, his3-Δ200, leu2-3,112 |
| Strain, strain background (*Saccharomyces cerevisiae*) | cdc23-1, SPC105-mScarlet-I, KIP1-GFP, HaloTag-TUB1 | This paper | DDY 5833 | MATa, cdc23-1, SPC105-mScarlet-I::LEU2, KIP1-GFP::KanMX, ura3-52::PHIS3-HaloTag-TUB1::URA3, his3-Δ200, leu2-3,112, lys2-801(am) |
| Strain, strain background (*Saccharomyces cerevisiae*) | cdc23-1, kip3 480 GCN4 coiled-coil-EYFP, SPC105-mScarlet-I, KIP1-GFP, HaloTag-TUB1 | This paper | DDY 5835 | MATalpha, cdc23-1, kip3Δ::KanR::pKIP3-kip3 480 GCN4 coiled-coil-EYFP-tADH::LEU2, SPC105-mScarlet-I::LEU2, his3-Δ200, leu2-3,112, lys2-801(am), ura3-52 |
| Strain, strain background (*Saccharomyces cerevisiae*) | cdc23-1, KIP2-GFP, SPC105-mScarlet-I, HaloTag-TUB1 | This paper | DDY 5836 | MATa, cdc23-1, KIP2-3x flexible linker-yoEGFP::KanMX, SPC105-mScarlet-I::LEU2, ura3-52::PHIS3-HaloTag-TUB1::URA3, his3-Δ200, leu2-3,112, lys2-801(am) |
| Strain, strain background (*Saccharomyces cerevisiae*) | cdc23-1, kip2Δ, SPC105-GFP, mRuby2-TUB1 | This paper | DDY 5837 | MATa, cdc23-1, kip2Δ::CgHIS3, SPC105-3x flex linker-yoEGFP::KanMX, TUB1::PHIS3-mRuby2-TUB1::HPH, his3-Δ200, leu2-3,112, lys2-801(am), ura3-52 |
| Strain, strain background (*Saccharomyces cerevisiae*) | cdc23-1, kip2Δ, SPC105-GFP, mRuby2-TUB1 | This paper | DDY 5838 | MATalpha, cdc23-1, kip2Δ::CgHIS3, SPC105-3x flex linker-yoEGFP::KanMX, TUB1::PHIS3-mRuby2-TUB1::HPH, his3-Δ200, leu2-3,112, lys2-801(am), ura3-52 |
| Strain, strain background (*Saccharomyces cerevisiae*) | cdc23-1, kip1Δ, SPC105-GFP, mRuby2-TUB1 | This paper | DDY 5840 | MATalpha, cdc23-1, SPC105-GFP::kanMX, TUB1::PHIS3-mRuby2-TUB1::HPH, KIP1Δ::HIS3M × 6, his3-Δ200, leu2-3,112, lys2-801(am), ura3-52 |
| Strain, strain background (*Saccharomyces cerevisiae*) | cdc23-1, kip1Δ, SPC105-GFP, mRuby2-TUB1 | This paper | DDY 5841 | MATa, cdc23-1, SPC105-GFP::kanMX, TUB1::PHIS3-mRuby2-TUB1::HPH, KIP1Δ::HIS3M × 6, his3-Δ200, leu2-3,112, lys2-801(am), ura3-52 |
| Strain, strain background (*Saccharomyces cerevisiae*) | cdc23-1, ASK1-GFP, mRuby2-TUB1 | This paper | DDY 5842 | MATalpha, his3Δ200, lys2-801(am) am, ura3-52, leu2-3,112, ASK1-EGFP::kanMX, TUB1::PHIS3-mRuby2-TUB1::HPH, cdc23-1 |

*Continued on next page*

*Continued*

| Reagent type (species) or resource | Designation | Source or reference | Identifiers | Additional information |
|---|---|---|---|---|
| Strain, strain background (*Saccharomyces cerevisiae*) | cdc23-1, ASK1-GFP, mRuby2-TUB1 | This paper | DDY 5843 | MATa, his3Δ200, lys2-801(am), ura3-52, leu2-3,112, ASK1-EGFP::kanMX, TUB1::PHIS3-mRuby2-TUB1::HPH, cdc23-1 |
| Strain, strain background (*Saccharomyces cerevisiae*) | cdc23-1, KAR3-AID, TIR1, SPC105-GFP, mRuby2-TUB1 | This paper | DDY 5844 | MATalpha, lys2, KAR3-AID-9myc::HIS3, TIR1::LEU2, SPC105-GFP::kanMX, TUB1::PHIS3-mRuby2-TUB1::HPH, cdc23-1, his3-Δ200, leu2-3,112, lys2-801(am), ura3-52 |
| Strain, strain background (*Saccharomyces cerevisiae*) | cdc23-1, KAR3-AID, TIR1, SPC105-GFP, mRuby2-TUB1 | This paper | DDY 5845 | MATalpha, lys2, KAR3-AID-9myc::HIS3, TIR1::LEU2, SPC105-GFP::kanMX, TUB1::PHIS3-mRuby2-TUB1::HPH, cdc23-1, his3-Δ200, leu2-3,112, lys2-801(am), ura3-52 |
| Strain, strain background (*Saccharomyces cerevisiae*) | cdc23-1, NDC80-GFP, mRuby2-TUB1 | This paper | DDY 5846 | MATalpha, his3Δ200, lys2-801(am), ura3-52, leu2-3,112, NDC80-EGFP::kanMX, TUB1::PHIS3-mRuby2-TUB1::HPH, cdc23-1 |
| Strain, strain background (*Saccharomyces cerevisiae*) | cdc23-1, NDC80-GFP, mRuby2-TUB1 | This paper | DDY 5847 | MATa, his3Δ200, lys2-801(am), ura3-52, leu2-3,112, NDC80-EGFP::kanMX, TUB1::PHIS3-mRuby2-TUB1::HPH, cdc23-1 |
| Strain, strain background (*Saccharomyces cerevisiae*) | cdc23-1, CNN1-GFP, mRuby2-TUB1 | This paper | DDY 5848 | MATalpha, his3Δ200, lys2-801(am), ura3-52, leu2-3,112, CNN1-EGFP::kanMX, TUB1::PHIS3-mRuby2-TUB1::HPH, cdc23-1 |
| Strain, strain background (*Saccharomyces cerevisiae*) | cdc23-1, CNN1-GFP, mRuby2-TUB1 | This paper | DDY 5849 | MATalpha, his3Δ200, lys2-801(am), ura3-52, leu2-3,112, CNN1-EGFP::kanMX, TUB1::PHIS3-mRuby2-TUB1::HPH, cdc23-1 |
| Strain, strain background (*Saccharomyces cerevisiae*) | cdc23-1, GFP-KAR3, SPC105-mScarlet-I | This paper | DDY 5850 | MATa, EGFP-KAR3::KanMX, SPC105-mScarlet-I::LEU2, cdc23-1, his3-Δ200, leu2-3,112, lys2-801(am), ura3-52 |
| Strain, strain background (*Saccharomyces cerevisiae*) | cdc23-1, MIF2-GFP, MTW1-mScarlet-I | This paper | DDY 5851 | MATalpha, MIF2-GFP::KanMX, MTW1-mScarlet-I::LEU2, cdc23-1, lys2-801(am), ura3-52, leu2-3,112, his3Δ200 |
| Chemical compound, drug | Benzonase, purity >90% | EMD Millipore | Cat. Num: 70746–3 | |
| Chemical compound, drug | Concavalin A | Sigma-Aldrich | Cat. Num: C2010 | |
| Chemical compound, drug | GMPCPP | Jena Bioscience | Cat. Num: NU-405 | |
| Chemical compound, drug | Hellmanex III | Hellma Analytics | Cat. Num: 9-307-011-4-507 | |
| Chemical compound, drug | 3-indole acetic acid, 98% | Sigma-Aldrich | Cat. Num: I3750 | |
| Chemical compound, drug | Janelia Fluor 646 HaloTag ligand | Promega | Cat. Num: GA1120 | |
| Chemical compound, drug | poly-L-lysine grafted with polyethylene glycol | SuSoS AG | PLL(20)-g[3.5]-PEG(2) | |
| Chemical compound, drug | poly-L-lysine grafted with polyethylene glycol-biotin (50%) | SuSoS AG | PLL(20)-g[3.5]-PEG(2)/PEG(3.4)-biotin(50%) | |
| Chemical compound, drug | protease inhibitor cocktail IV | Calbiochem | Cat. Num: 539,136 | |
| Other | BSA gold tracer, 10 nm | Electron Microscopy Science | Cat. Num: 25,486 | Used for electron microscopy |
| Other | holey carbon electron microscopy grid, Quantifoil R 2/1, copper grid | Electron Microscopy Science | Cat. Num: Q210CR1 | Used for electron microscopy |
| Recombinant protein | Tubulin protein (HiLyte Fluor 647) - Porcine brain | Cytoskeleton | Cat. Num: TL670M | |
| Recombinant protein | Tubulin protein (Biotin) - Porcine brain | Cytoskeleton | Cat. Num: T333P | |
| Software, algorithm | Fiji/ImageJ | NIH | | https://imagej.net/ |
| Software, algorithm | IMOD | University of Colorado | | https://bio3d.colorado.edu/imod/ |
| Software, algorithm | Prism | Graphpad | | |

## Yeast strains, culturing, and harvesting

Yeast strains used in this study can be found in the above Key Resources Table. Fluorescent tagging, degradation tags, and deletions were integrated as previously described (*Longtine et al., 1998*; *Lee et al., 2013*; *Bindels et al., 2017*). All strains were grown in standard rich medium (YPD) at 25°C since they contained a temperature-sensitive allele.

Cells were harvested as previously described in *Bergman et al., 2018*. In brief, strains were grown overnight in starter cultures and then diluted back into two identical 2 L cultures of YPD at an OD600=6.25 × 10$^{-3}$. At an OD600=0.4 cultures were then shifted to 37°C for a 3 hr arrest. Strains with degradation tags were treated with 250 µM 3-indole acetic acid (Sigma-Alrich, St. Louis, MO) in DMSO and 50 mM potassium phosphate buffer, pH 6.2, for 30 min before harvesting, 2.5 hr after the temperature shift. Cells were then harvested by serial centrifugation at 6000 x g in a Sorvall RC5B centrifuge using an SLA-3000 rotor for 10 min at 4°C. Cell pellets were then resuspended in cold ddH2O and pelleted in a Hermle Z446K centrifuge for 3 min at 3430 x g at 4°C. This wash procedure was repeated once more, with any remaining liquid removed from the pellet via aspiration. Using a stainless steel lab spatula (Thermo Fisher, Waltham MA), the cells were then scraped out as small clumps into a 50 mL conical filled with liquid nitrogen, snap-freezing them, for storage at -80°C.

## Generation of whole cell lysates

As described previously by *Bergman et al., 2018*, approximately 5 g of frozen cells were weighed into a pre-chilled medium-sized SPEX 6870 freezer mill vial (Spex, Metuchen, NJ). The pre-chilled chamber was then milled (submerged in liquid nitrogen) following a protocol that consisted of a 3 min pre-chill, then 10 cycles of 3 min grinding at 30 impacts per second (15 cps) and 1 min of rest. The resulting powered lysate was stored at -80°C.

## Generating stabilized far-red labeled tubulin seeds

As previously described in *Bergman et al., 2018*, purified bovine tubulin was cycled to remove nonfunctional tubulin. This tubulin was then mixed with both biotin-conjugated and HiLyte 647 labeled porcine tubulin (Cytoskeleton Inc, Denver, CO). The tubulin mix was then resuspended in PEM buffer (80 mM PIPES pH 6.9, 1 mM EGTA, 1 mM MgCl2) to a final concentration of 1.67 mg/mL unlabeled tubulin, 0.33 mg/mL biotin-labeled tubulin, and 0.33 mg/mL far-red labeled tubulin. To stabilize the seeds, GMPCPP (Jena Biosciences, Jena, Germany) was added to a final concentration of 1 mM. Aliquots were then snap-frozen in liquid nitrogen and stored at -80°C.

## Preparation of glass sides and passivation of coverslips

Both glass side preparation and passivation of coverslips followed the protocol previously described in *Bergman et al., 2018*. To prepare the glass sides for assembly into a flow chamber, microscope slides (Corning Inc, Corning, NY) were washed in acetone for 15 min and then 100% ethanol for 15 min. These slides were left to air dry before storage in an airtight container.

To prepare the coverslips for assembly into a flow chamber, cover glass (1.5 thickness, Corning Inc) was first cleaned by sonication in acetone for 30 min. The coverslips were then soaked for 15 min in 100% ethanol. After 3 thorough, but short, rinses in ddH$_2$O, the coverslips were submerged for 2 hr in 2% Hellmanex III solution (Hellma Analytics, Müllheim, Germany). After this, they were again rinsed in ddH2O three times. Before proceeding to passivation, the coverslips were blown dry with nitrogen gas. For passivation, a solution containing a 0.1 mg/mL mixture of PLL(20)-g[3.5]-PEG(2):PEG(3.4)-biotin(50%) (SuSoS AG, Dübendorf, Switzerland), at a ratio of 1:19, in 10 mM HEPES was prepared. Fifty µL drops were then placed on Parafilm in a humid chamber and coverslips were gently placed onto the drops. After 1 hr, the passivated coverslips were washed for 2 min in PBS and rinsed in ddH2O for 1 min. The coverslips were then air-dried with nitrogen gas and stored in an airtight container at 4°C. These passivated coverslips were stored for use only for three weeks.

## Assembly of flow chamber

Preparation and assembly of the flow chamber for use in the TIRF based dynamics assay was exactly as described in *Bergman et al., 2018*.

## Preparation of whole cell lysates for dynamics assay

Similarly to what is described in *Bergman et al., 2018*, 0.22 g of powdered lysate was weighed out into a 1.5 mL Eppendorf tube pre-chilled in liquid nitrogen. To assemble microtubules of a physiologically desired length (5 µm), a range of 1–25 µL of cold 10 X PEM (800 mM PIPES pH 6.9,10 mM MgCl2, 10 mM EGTA) was added to the powered lysate, as some strains required more or less buffer volumes to obtain 5 µm long microtubules in the assay. In addition to cold 10 X PEM, 0.5 µL of Protease Inhibitor Cocktail IV (Calbiochem, San Diego, CA) and 4 µL of 25.5 U/µL benzonase nuclease (EMD Millipore, San Diego, CA, prod. 70746–3,>90% purity) was added, spun down briefly, and thawed on ice for 10 min. Lysate was added to pre-chilled polycarbonate ultracentrifuge tubes and cleared of insoluble material by spinning at 34,600 x g for 25 min at 4°C. After ultracentrifugation, 32 µL of cleared lysate supernatant was flowed into the chamber prepared previously (see above).

## TIRF microscopy

After clarified lysate supernatant was added to the prepared chamber, the slides were loaded onto a Nikon Ti2-E inverted microscope with an Oko Labs environmental chamber pre-warmed to 28°C. Images were acquired using a Nikon 60 X CFI Apo TIRF objective (NA 1.49) and an Orca Fusion Gen III sCMOS camera (Hamamatsu, Hamamatsu City, Japan) at ×1.5 magnification using the Nikon NIS Elements software. Using a LUNF 4-line laser launch (Nikon Instruments, Melville, NY) and an iLas2 TIRF/FRAP module (Gataca Systems, Massy, France) total internal reflection fluorescence (TIRF) illuminated a single focal plane of the field and was imaged every 5 s for 30 min.

## Sequential lysate flow assays

Similar coverslip, slide, assembly of flow chamber, and lysate preparation were all followed as described above. However, instead of preparing one lysate for one genotype, two lysates were prepared side-by-side. To observe the initial behavior, 32 µL of the first lysate was flowed through the chamber and imaged for 10 min. While the slide was still mounted on the microscope, the movie was paused for 1 min while the second lysate was carefully flowed in, replacing the first. After replacing the lysate, the movie was restarted and the same field of view was captured for another 20 min.

## In vivo live cell microscopy

Cells were grown in rich medium (YPD) overnight to saturation at 25. The next day the cells were diluted into fresh YPD and grown for 4 hr (2 doublings) at 25°C, until the culture reached log phase (OD600=0.5). They were then spun down and washed with minimal Imaging Medium three times (synthetic minimal medium supplemented with 20 µg/ml adenine, uracil, L-histidine and L-methionine; 30 µg/ml L-leucine and L-lysine; and 2% glucose; Sigma-Aldrich). Cells were then immobilized on coverslips coated with 0.2 mg/ml concanavalin A and were imaged in Imaging Medium. Using a Nikon Ti2-E inverted microscope with an Oko Labs environmental chamber pre-warmed to 25°C. Images were acquired with a Nikon 60 X CFI Apo TIRF objective (NA 1.49) and an Orca Fusion Gen III sCMOS camera (Hamamatsu) at 1.5 X magnification using the Nikon NIS Elements software. For imaging, a LUNF 4-line laser launch (Nikon) and an iLas2 TIRF/FRAP module (Gataca Systems) was used for HiLo total internal reflection fluorescence (*Tokunaga et al., 2008*). Images were taken with 0.2 µm slices for a total 5 µm Z-stack.

## Image and data analysis

For the microtubule dynamics assay, analysis was done as described previously in *Bergman et al., 2018*. Imaging data were analyzed using Fiji software (NIH). Registration to correct for stage drift was applied to the raw data (StackReg; *Thévenaz et al., 1998*). Kymographs were generated from all microtubules for which the entire length could be tracked for the entire movie. Kymographs were excluded if the microtubules were crossed or bundled. For analysis, data from independent technical trials and biological replicas from one genotype were pooled, unless otherwise indicated. In our quantification of speeds, a minimum threshold of 86.7 nm/min for kinetochore movement rates was established based on the resolution of the kymographs used to measure speed. Kinetochore movements less than a threshold slope of 1pixel displacement (72.2 nm) per 10 pixels of time (50 s) were categorized as 'paused'. Velocities are reported as mean with the standard error of the mean.

Statistical significance was determined using a Kruskal-Wallis test (GraphPad Prism, San Diego, CA). p Values are reported as in the figure captions.

For the colocalization quantification, pair-wise kinetochore tracks (that moved as defined above) were pooled into three categories (always, partially, and never colcalized) based on the amount of time that the two proteins were colocalized. Because the red fluorophore blinked, colocalization was scored solely for the mScarlet tagged protein (either Ndc80 or Mtw1).

For the live cell imaging analysis, images were analyzed using Fiji (NIH). Maximum intensity Z-projections were made and bleach corrections were applied using the 'histogram matching' macro. Metaphase cells were identified by the presence of a 2–3 µm spindle at the entrance to the bud neck. Cells were then counted as either 'bilobed' or 'declustered' based on the presence of two distinct kinetochore puncta. Line scans were done to assist in this binary classification. If two clear peaks were present, the cell was 'bilobed'. Any other line scan shape, that is, 3+peaks or 1 long kinetochore signal that matched the spindle, were classified as 'declustered'. Data are from two independent technical trials of one biological replicate. In each replicate, n=50 cells were counted. Graphs are of the mean and standard error of the mean (GraphPad Prism).

## Cryo-sample preparation

Holey carbon electron microscopy grids (Quantifoil R2/1, 200 mesh, copper; Electron Microscopy Science, Hatfield, PA) were washed in 99.5% acetone while agitating on a rocker for 15–30 min to remove potential residual plastic backing, then washed in H2O, and dried with filter paper. Cell lysate of DDY5818 (cdc23-1, SPC105-3xFlexLinker-yoEGFP, mRuby2-TUB1) strain was prepared as described above. After ultracentrifugation, supernatants were aliquoted into pre-chilled microcentrifuge tubes with a volume of 9.5 µL supernatant per aliquot and kept on ice. About 15 min before the lysate centrifugation was finished, far-red labeled GpCpp-stabilized MT seeds were polymerized in microcentrifuge tubes at 37°C for 10 min. Grids were coated with 10 nm BSA Gold Tracer (Electron Microscopy Sciences) by dripping a total of 10–20 µL of the BSA Gold Tracer solution on the grids and removing the drops with filter paper. Seeds were diluted 1/50 in 1 X PEM buffer. 0.5 µL of seeds were mixed with 9.5 µL cleared lysate. Three to 6 µL of the mix were pipetted on the pre-coated grids. For sample vitrification, either a Vitrobot Mark IV (FEI, Hillsboro, OR) or a Leica EM GP2 (Leica Microsystems, Wetzlar, Germany) was used. Sample chamber conditions were set to 28°C and 75–90% humidity. Samples were incubated in the sample chamber for 10–15 min and then blotted from the backside and vitrified by plunging into liquid ethane. To set up the Vitrobot Mark IV for back-blotting, the filter paper facing the sample side was replaced with a custom-made Teflon sheet of the same size to blot only the fluid from the back of the sample. To ensure that suitable grids were produced for each imaging session, a range of plot forces between 0 and 10 (for Vitrobot Mark IV only), blotting times of 3–6 s, and a 1 s drain time, were used. Grids were fixed into AutoGrid carriers (Thermo Fisher) and stored in liquid nitrogen until they were imaged.

## Cryo-fluorescence light microscopy

Cryo-samples were imaged on a Leica EM Cryo CLEM system (Leica Microsystems). The system consists of a Leica DM6 FS widefield microscope that is equipped with a motorized Leica EM Cryo stage and a short working distance (<0.28 mm) 50 x Leica EM Cryo CLEM ceramic tipped objective (numerical aperture = 0.90). These specifications allow sample imaging at liquid nitrogen temperatures. A halogen lamp powered by a CTR6 halogen supply unit was used as a light source. We used GFP ET (Excitation: 470/40, Dichroic: 495, Emission 525/50), RFP (Excitation: 546/10, Dichroic: 560, Emission 585/40) and Y5 (Excitation: 620/60, Dichroic: 660, Emission 700/75) filter cubes for imaging. For cryo-correlative light and electron microscopy, a grid overview map was recorded using transmitted light and Y5 or GFP channels. The map was used to identify grid squares with good ice quality and intact carbon film. Z-stacks (total Z=7 µm in 0.5 µm steps) of regions with clear fluorescence signals were recorded using the transmitted light, GFP (Spc105), RFP (Tub1), and Y5 (seeds) channels. These images were used to identify Spc105-GFP-positive regions of interest for subsequent cryo-electron tomography imaging. The same imaging conditions were used for grid squares on individual grids. Imaging conditions were varied between grids to obtain sufficient signal to allow correlation in later steps. Images for panel generation in Adobe Illustrator were prepared using Fiji and Adobe Photoshop. No nonlinear gamma correction was applied during image processing.

## Cryo-electron tomography data acquisition and tomogram reconstruction

Samples were imaged on a Titan Krios transmission electron microscope (FEI) equipped with an X-FEG electron source, a BioQuantum energy filter (Gatan, Pleasanton, CA) and a K3 direct electron detecting device (Gatan) and operated at 300 kV. Samples were visually inspected for ice quality using the FEI flu cam. Overview grid maps were acquired at 0.2 µm pixel size. These grid maps were used to identify grid squares of interest from the cryo-fluorescence light microscopy overview maps. Grid square images and polygon maps of the regions of interest with an overlap of 20–25% between individual images were recorded. The polygon maps were used to pick tilt series acquisition points based on the fluorescent signal from cryo-fluorescence light microscopy. The hole pattern of the carbon film was used as a guide. Acquisition points were chosen with adequate distance between individual points to prevent electron dose exposure damage prior to data collection. SerialEM (*Mastronarde, 2005*) in low-dose mode was used for automatic tilt-series recording using a bidirectional tilt scheme starting at +20°and with a typical tilt range from +60°to –60°and base increment of 2°. Pixel sizes between 2.63 and 3.07 Å were used. Target defocus was varied between –2 and -6 µm and the targeted total electron dose was 100 $e^-/Å^2$. Data were collected in 0.5 binning mode or without binning. Frame time was 0.2–0.25 s. Tilt series alignment and tomogram reconstruction were done using the freely available IMOD software package (*Kremer et al., 1996*). Tilt series were aligned using 10 nm gold particles as fiducials and tomograms were reconstructed using the backprojection algorithm combined with the SIRT-like filter. Tomograms were then filtered using the Nonlinear Anisotropic Diffusion filter and binned by a factor of 3 or 4 using the binvol function in IMOD to further increase contrast for analysis and visualization. Images for panel generation in Adobe Illustrator were prepared using IMOD and Adobe Photoshop. No nonlinear gamma correction was applied during image processing.

## Acknowledgements

The authors thank the Pellman lab and Xiaolei Su for Kip3 mutant allele constructs and valuable help. We would also like to thank Zane Bergman and Jennifer Hill for advice and support. Additionally, thank you to Doug Koshland for reading the manuscript and providing valuable edits. This work was supported by the National Institutes of Health (Grant R01 GM 47842) to G.B., funds from the Judy Chandler Webb Endowed Chair in the Biological Sciences to G.B. and a Human Frontier Science Program long term fellowship (LT000234/2018 L) to D.S.

## Additional information

### Funding

| Funder | Grant reference number | Author |
| --- | --- | --- |
| National Institutes of Health | Grant R01 GM 47842 | Georjana Barnes |
| Judy Chandler Webb Endowed Chair in the Biological Sciences | | Georjana Barnes |
| Human Frontier Science Program | LT000234/2018-L | Daniel Serwas |

The funders had no role in study design, data collection and interpretation, or the decision to submit the work for publication.

### Author contributions

Julia R Torvi, Conceptualization, Data curation, Formal analysis, Investigation, Methodology, Validation, Visualization, Writing – original draft, Writing – review and editing; Jonathan Wong, Conceptualization, Data curation, Formal analysis, Investigation, Methodology, Validation, Visualization, Writing – review and editing; Daniel Serwas, Formal analysis, Funding acquisition, Investigation, Methodology, Resources, Validation, Visualization, Writing – review and editing; Amir Moayed, Formal analysis,

Visualization; David G Drubin, Project administration, Resources, Supervision, Writing – review and editing; Georjana Barnes, Funding acquisition, Project administration, Resources, Supervision, Writing – review and editing

### Author ORCIDs
Julia R Torvi ![ORCID] http://orcid.org/0000-0002-2323-4438
Jonathan Wong ![ORCID] http://orcid.org/0000-0002-1973-0156
Daniel Serwas ![ORCID] http://orcid.org/0000-0001-9010-7298
David G Drubin ![ORCID] http://orcid.org/0000-0003-3002-6271
Georjana Barnes ![ORCID] http://orcid.org/0000-0003-2083-358X

### Decision letter and Author response
Decision letter https://doi.org/10.7554/eLife.78450.sa1
Author response https://doi.org/10.7554/eLife.78450.sa2

## Additional files

### Supplementary files
• Transparent reporting form

### Data availability
All data analysed during this study are included in the manuscript and supporting file; Source Data files have been provided for Figures 1, 4, and 5. This includes both the numerical data use to generate the figures, and the graphs containing statistical tests done.

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
