## [Editor Report]

Kinetochores are large protein complexes that mediate faithful chromosome segregation in eukaryotes. The authors develop an in vitro approach to study interactions between kinetochores and microtubules in yeast cell extracts. They use this powerful lysate-based system to characterize a new role for the budding yeast kinesin-8, Kip3, in powering lateral kinetochore movement along microtubules. This paper should be of interest to researchers working in the field of mitosis, the cell cycle, and the cytoskeleton, and, more broadly, for those studying macromolecular complexes with reconstitution and in vitro imaging approaches.

---

## [Decision Letter]

**Decision letter after peer review:**

Thank you for submitting your article "Reconstitution of Kinetochore and Microtubule Dynamics Reveals a Role for a Kinesin-8 in Establishing End-on Attachments" for consideration by *eLife*. Your article has been reviewed by 3 peer reviewers, one of whom is a member of our Board of Reviewing Editors, and the evaluation has been overseen by Anna Akhmanova as the Senior Editor. The following individuals involved in review of your submission have agreed to reveal their identity: Jason Stumpff (Reviewer #2); Stefan Westermann (Reviewer #3).

Essential revisions:

1. An interesting observation is that Kip3 doesn't consistently co-localize with the kinetochore particles, which raises a question regarding the mechanism of Kip3-mediated kinetochore movement. The authors suggest that weak transient interactions with Kip3 facilitate kinetochore movement or alternatively, that Kip3 is indeed stably associated with kinetochores, but low stoichiometry prevented detection in light of high background of free Kip3 bound to microtubules. To address this, the authors should consider titrating in recombinant Kip3 into Kip3 deficient extracts – this may allow for detection of a kinetochore-bound population of Kip3, and will reveal how much Kip3 is required to initiate kinetochore movement in this system.

2. It is not clear if the authors are arguing for a model in which: (1) Kip3 travels to the plus end, where it causes depolymerization, which then allows for end-on conversion; or (2) Kip3 travels to the plus end, where it converts kinetochore attachments from lateral to end-on, which then causes depolymerization (ex, Lines 211-215; Lines 251-253; lines 270-273; Lines 288-289; Lines 306-307). To help clarify this, the authors should report the following for the WT, Kip3-depleted, and Kip3 mutant conditions: In what percentage of cases where kinetochores did arrive to the MT end, did this cause depolymerization (and how does this compare to the frequency of spontaneous depolymerization without KT arrival)?

3. In Figure 4e, the authors show images suggesting that adding exogenous Kip3 to Kip3-depleted lysates (in which kinetochores have assembled and bound to microtubules) promotes translocation of the kinetochore particles along the microtubules. Please provide quantification for this, so that it is clear if these are rare or common events.

4. Please provide quantification for the data in Figure 2.

5. The kinetochore densities in the tomographic slices shown in Figure 3b and 3d are not easy to see in the Figure, and are only a bit easier to see in the supplemental movies. It may be helpful to add another panel of the same images with the kinetochores outlined to aid with visualization.

6. In Figure 1, is there an explanation for why the plus-end directed velocities of kinetochores with fluorescent labels on Mif2 and Mtw1 move significantly slower than those with other labels? The Mtw1-labeled tip-bound kinetochores also seem to spend more time in minus-end directed movement than the other conditions. Is this expected? Please address this in the text.

7. On Line 174, the authors state "Interestingly, although less frequent and covering shorter distances, kinetochore movements in kip3∆ lysates had the same velocity as the more frequent and longer movements observed in Kip3+ lysates." However, in the discussion on Line 310, the authors state "Kinetochores did not move on microtubule in the kip3∆ lysate." The apparent contradiction between these statements should be addressed. It would also be interesting to know if the authors have thoughts on how kinetochores may be moving directionally in the absence of Kip3. Is there an intrinsic mechanism that promotes kinetochore movement towards the microtubule plus-end? If so, has this been observed in previous in vitro studies of kinetochores?

8. The data in Figure 4b appear to indicate that loss of cin8 activity increases the percentage of time that tip-bound kinetochores spend in minus-end directed movement. Is this a significant increase and, if so, is there a logical explanation for this effect? Cin8 has been proposed to promote disassembly of kinetochore microtubule plus-ends (PMID: 19041752).

9. Please provide in the discussion a comparison of this study to the Tanaka re-activated CEN system (Tanaka et al., Nature 2005), which is still widely considered the gold standard in the field to directly observe different attachment configurations in yeast cells. In this system, no motor except Kar3 showed an effect. This should be addressed in the text.

10. Please provide a brief discussion regarding conceptual limitations of these lysates: Kip3 has roles in nucleus and cytoplasm, yet this distinction is lost in these whole cell lysates. This is especially critical for budding yeast which undergoes a closed mitosis. What are the prospects of using nuclear extracts instead?

*Reviewer #1 (Recommendations for the authors):*

Specific Comments:

(1) In figure 3D, the authors note a "cloud-like density" at the tips of microtubules. While I see this density in 3b, it is not obviously present in 3d. Is it possible to provide clearer images?

(2) In Figure 4e, the authors show images suggesting that adding exogenous Kip3 to Kip3-depleted lysates (in which kinetochores have assembled and bound to microtubules) promotes translocation of the kinetochore particles along the microtubules. Providing quantification of this would help the reader understand if this a rare or common event.

(3) On line 202, the authors state that only Kip3 and Kip2 moved processively on the microtubules, and the callout indicates "Figure 5," but I don't see the data.

(4) In many instances it is unclear as to whether the authors are suggesting that Kip3 functions to decrease or increase microtubule dynamics. They state in several cases that Kip3 attenuates microtubule dynamics (lines 56, 62, 239, 306) and in other instances they mention that Kip3 functions to increase dynamics (lines 49, 183, 230-231). For a reader not familiar with the details of the Kinesin-8 family of motors, this is somewhat confusing. It would be helpful to provide some clarity here.

*Reviewer #2 (Recommendations for the authors):*

Overall, I think this is a well-executed study with conclusions that are supported by high quality data. I do have a few minor suggestions for the authors to consider that might improve the manuscript.

1. In Figure 1, is there an explanation for why the plus-end directed velocities of kinetochores with fluorescent labels on Mif2 and Mtw1 move significantly slower than those with other labels? The Mtw1-labeled tip-bound kinetochores also seem to spend more time in minus-end directed movement than the other conditions. Is this expected?

2. On Line 174, the authors state "Interestingly, although less frequent and covering shorter distances, kinetochore movements in kip3∆ lysates had the same velocity as the more frequent and longer movements observed in Kip3+ lysates." However, in the discussion on Line 310, the authors state "Kinetochores did not move on microtubule in the kip3∆ lysate." The apparent contradiction between these statements should be addressed. It would also be interesting to know if the authors have thoughts on how kinetochores may be moving directionally in the absence of Kip3. Is there an intrinsic mechanism that promotes kinetochore movement towards the microtubule plus-end? If so, has this been observed in previous in vitro studies of kinetochores?

3. The data in Figure 4b appear to indicate that loss of cin8 activity increases the percentage of time that tip-bound kinetochores spend in minus-end directed movement. Is this a significant increase and, if so, is there a logical explanation for this effect? Cin8 has been proposed to promote disassembly of kinetochore microtubule plus-ends (PMID: 19041752).

4. The kinetochore densities in the tomographic slices shown in Figure 3b and d are not easy to see in the figure and are only a bit easier to see in the supplemental movies. It may be helpful to add another panel of the same images with the kinetochores outlined to aid with visualization.

5. For the histograms shown in 4 and 5, changing the y-axis to percent of total may make it easier to compare the data sets displayed on each graph.

*Reviewer #3 (Recommendations for the authors):*

I think in principle this is a well-developed study and I hesitate to ask for many more experiments. I think in principle this is suitable for *eLife*.

Given that the mechanism of Kip3-mediated kinetochore movement is still somewhat unclear, it would be interesting to titrate in recombinant Kip3 into the Kip3 deficient extracts, and not just perform the complementation with Kip3 containing extracts. This might facilitate the detection of a Kinetochore-bound population of Kip3 or determine how much Kip3 is required to initiate kinetochore movement in this system.

To improve data presentation and discussion I would further suggest the following points:

1) Some more quantification of data on key observations: Does the arrival of a kinetochore at a plus-end really switch that microtubule to depolymerization? In some of the presented kymographs it looks to me like the depolymerization initiates before the arrival (e.g. 1c, Ndc80 example). What is the fraction of arrival events that lead to depolymerization versus the frequency of spontaneous depolymerization without KT arrival? Also, in Figure 2 some more quantification on the co-localization would be helpful.

2) The discussion should include two additional points:

Comparison to the Tanaka re-activated CEN system (Tanaka et al., Nature 2005, which is still the gold standard in the field to directly observe different attachment configurations in yeast cells). In this system no motor except Kar3 showed an effect.

Also, a brief discussion regarding conceptual limitations of these lysates: Kip3 has roles in nucleus and cytoplasm, yet this distinction is lost in these whole cell lysates. This is especially critical for budding yeast which undergoes a closed mitosis. What are the prospects of using nuclear extracts instead?

---

## [Author Response]

Essential revisions:1. An interesting observation is that Kip3 doesn't consistently co-localize with the kinetochore particles, which raises a question regarding the mechanism of Kip3-mediated kinetochore movement. The authors suggest that weak transient interactions with Kip3 facilitate kinetochore movement or alternatively, that Kip3 is indeed stably associated with kinetochores, but low stoichiometry prevented detection in light of high background of free Kip3 bound to microtubules. To address this, the authors should consider titrating in recombinant Kip3 into Kip3 deficient extracts – this may allow for detection of a kinetochore-bound population of Kip3, and will reveal how much Kip3 is required to initiate kinetochore movement in this system.

To address the possibility that unbound Kip3 obscures the visibility of Kip3 associated with kinetochores, the reviewers state, “the authors should consider titrating in recombinant Kip3 into Kip3 deficient extracts”. We appreciate this suggestion, which we attempted to act on as described below, and which motivated us to add a new passage into the Discussion.

Unfortunately, we are currently unable to purify functional Kip3 and add it to the extract assay without diluting the extract and completely losing all extract activity. However, in response to this reviewer’s comment, we performed two experiments that utilized the same principle in an attempt to address the suggestion of titrating Kip3 into the assay.

While these two types of experiments (fully described below) failed to generate persistently colocalized tracks of moving kinetochores and Kip3, they do not rule out the possibility that Kip3 is present below the threshold of detection on motile kinetochores. However, these new negative results encouraged us to entertain alternative mechanisms for Kip3’s action at the end of the Discussion:

“In the absence of the ability to consistently observe direct Kip3 stable association with motile kinetochores, we cannot rule out other mechanisms to explain the Kip3-dependent kinetochore motility that we discovered. Further investigation is required to determine the exact molecular mechanism by which Kip3 mediates plus end-directed kinetochore transport on the microtubule lattice. It is possible, for example, that Kip3 transports an unidentified factor to kinetochores, and that this hypothetical factor mediates kinetochore movement.”

**Author response image 1. sa2fig1:** 

The goal of both experiments we tried was to “titrate” a labeled Kip3 protein into our assay, but in two different ways. In each experiment the goal was also to maintain extract assay microtubule functionality by avoiding dilution of overall cellular extract protein concentration. In the first experiment, we made a strain that expressed Kip3-HaloTag. By limiting the amount of dyeligand conjugate added to our lysates, we could control the total amount of fluorescent Kip3 visible on microtubules in the assay and could then look specifically for Kip3 tracks that colocalize with kinetochores. With the addition of JF646 ligand (Janelia Farms) to 25, 50, or 100 nM amounts, individual Kip3 tracks were easily distinguished. However, analysis of ~450 kymographs from lysates of metaphase-arrested cells did not reveal any labeled Kip3 colocalizing with slower moving Spc105-GFP (which marks kinetochores). A representative kymograph is shown to the right. One possible reason why this attempt might have failed is that unlabelled Kip3 might outcompete Kip3-Halo for binding to kinetochores, especially if the kinetochore was already assembled and had incorporated unlabeled Kip3 before the labeled Kip3 was added. Furthermore, in this experiment, limiting amounts of JF646 dye might not label Kip3-Halo already bound to kinetochores as efficiently as unbound Kip3-Halo, and using more dye to force binding would only increase the Kip3 background issue.In the second experiment, we titrated lysate from a Kip3-GFP strain with lysate from a *kip3∆* strain to control the amount of motor protein in our assay. At low concentrations of Kip3-GFP lysate, we did not observe a sufficient number of microtubule-associated kinetochores to analyze. However, increasing the amount of Kip3-GFP lysate in the mixture resulted in a dense population of motors on microtubules that obstructed any observation of colocalization with motile kinetochores, similar to what was observed in Figure 5c. Unfortunately, we were unable to achieve an optimal condition where we could have an unobstructed view of both motile side-bound kinetochores and individual Kip3 motors simultaneously.

2. It is not clear if the authors are arguing for a model in which: (1) Kip3 travels to the plus end, where it causes depolymerization, which then allows for end-on conversion; or (2) Kip3 travels to the plus end, where it converts kinetochore attachments from lateral to end-on, which then causes depolymerization (ex, Lines 211-215; Lines 251-253; lines 270-273; Lines 288-289; Lines 306-307). To help clarify this, the authors should report the following for the WT, Kip3-depleted, and Kip3 mutant conditions: In what percentage of cases where kinetochores did arrive to the MT end, did this cause depolymerization (and how does this compare to the frequency of spontaneous depolymerization without KT arrival)?

We thank the reviewers for pointing out the alternative way our statements could be interpreted. In order to clear up any ambiguity in what we believe the data indicates, we made the following changes.

Figure 5c text box: Changed “Percent of kinetochores that transition from side to end-bound” to “Percent of side-bound kinetochores that reached the microtubule plus end”.

Lines 234-240: Reworded several sentences to more clearly indicate support for model (1) and to avoid misinterpretation of our words as supporting model (2). We are proposing two specific roles for Kip3: (1) facilitating movement of kinetochores along the lateral surface of microtubules toward the plus-end and (2) induction of microtubule catastrophe triggered by Kip3, which decreases the distance required for laterally moving kinetochores to reach the microtubule tip. With respect to the two models put forth above by the reviewers, we propose that model (1) is correct because fewer side-bound kinetochores catch up to the microtubule end in a lysate made from a *kip3∆T-LZ* (12.4%) strain because this Kip3 mutant has impaired ability to induce catastrophe (as indicated in Figure 5c). However, this mutant retains the ability to move kinetochores along microtubules. Since few kinetochores reach the microtubule plus-end in *kip3∆T-LZ or kip3∆* lysates, a comparison of the behaviors of kinetochore-associated microtubule ends between lysates of mutant and wild-type strains be of limited value because it would be very hard to achieve statistical significance due to low frequency of kinetochores associating with MT plus ends. Nevertheless, we did attempt to quantify the behaviors and found the following:

**Author response table 1. sa2table1:** Immediate Behavior of Kinetochores that Reach the Microtubule Plus-End.

	Wild-type Kip3	*kip3∆*	Kip3∆T-LZ
Move toward the seed	27 (75%)	4 (57%)	3 (30%)
No net movement	9 (25%)	3 (43%)	7 (70%)
Move away from the seed	0 (0%)	0 (0%)	0 (0%)

While the decrease in the percentage of kinetochores that track depolymerizing microtubules in lysates from *kip3* mutants immediately after they reach the microtubule tip is consistent with the possibility that Kip3 associated with end-bound kinetochores might induce depolymerization, we do not want to propose this idea in the manuscript because the observable sample size was too small and other possible explanations exist.

With respect to model (2) proposed by the reviewers, while the Kip3 motor protein may facilitate the transition of kinetochores from a side-bound to end-bound orientation on microtubules, our current data neither supports nor refutes this possibility. The small number of events in which kinetochores reached the microtubule plus-end in lysates from *kip3* mutant strains made it difficult to study Kip3’s role directly. We are currently trying to design new strategies to monitor the transition of side-bound to end-bound kinetochores in vitro to test which proteins are essential for the process.

3. In Figure 4e, the authors show images suggesting that adding exogenous Kip3 to Kip3-depleted lysates (in which kinetochores have assembled and bound to microtubules) promotes translocation of the kinetochore particles along the microtubules. Please provide quantification for this, so that it is clear if these are rare or common events.

The text has been updated to indicate that 53.7% of stationary kinetochores in *kip3∆* lysates became motile (as defined by previous quantifications of motility in the Materials in Methods) once Kip3+ lysate was added, best shown in the representative kymograph in Figure 4e.

4. Please provide quantification for the data in Figure 2.

Quantification for this figure has been added and the figure legend has been updated along with the Materials and methods.

5. The kinetochore densities in the tomographic slices shown in Figure 3b and 3d are not easy to see in the Figure, and are only a bit easier to see in the supplemental movies. It may be helpful to add another panel of the same images with the kinetochores outlined to aid with visualization.

We have added another panel to Figures 3b and d with the densities colored in, as suggested.

6. In Figure 1, is there an explanation for why the plus-end directed velocities of kinetochores with fluorescent labels on Mif2 and Mtw1 move significantly slower than those with other labels? The Mtw1-labeled tip-bound kinetochores also seem to spend more time in minus-end directed movement than the other conditions. Is this expected? Please address this in the text.

We appreciate this comment and the opportunity to provide the following explanations and to modify our manuscript. In Figure 1, Mif2- and Cse4-labeled kinetochores move more slowly than kinetochores in which other proteins were tagged with fluorescent proteins. This observation is now addressed in the Results. These observations are consistent with the possibility that c-terminal GFP tags on Mif2 and Cse4 affect protein function and therefore dynamics on the microtubule. Both of the corresponding strains have slow growth rates. In addition, we had to make the linker for the GFP tag on Cse4 longer or else it was synthetic lethal with *cdc23-1* (doi: 10.1016/j.addr.2012.09.039). Despite the difference in measured lateral movement rates when these two kinetochore proteins were tagged, we believe that Figure 2 conclusively shows that Mif2- and Cse4- colocalize and move together with other kinetochore proteins.

With respect to Mtw1, while the speed was not significantly different from what was observed when kinetochore proteins other than Mif2 or Cse4 were tagged, kinetochores with tagged Mtw1 spend more time in minus-end directed movement when bound to the end of a microtubule than kinetochores in which other subunits were tagged. Whether this reflects a modification of Mtw1 function due to the fluorescent tag or some other factor whose nature is currently a mystery is unclear. However, the colocalization experiment in Figure 2 rules out the possibility that Mtw1 has a localization independent of other kinetochore proteins.

Taken together, our results indicate that the kinetics of microtubule-bound kinetochores in lysates is sensitive to perturbations in protein function and activity, but also that functional kinetochores all exhibit the same baseline characteristics of binding and plus-end directed movement on the lateral surface of microtubules followed by end-binding and tracking the depolymerizing ends of microtubules. We have modified the manuscript text to highlight these issues.

7. On Line 174, the authors state "Interestingly, although less frequent and covering shorter distances, kinetochore movements in kip3∆ lysates had the same velocity as the more frequent and longer movements observed in Kip3+ lysates." However, in the discussion on Line 310, the authors state "Kinetochores did not move on microtubule in the kip3∆ lysate." The apparent contradiction between these statements should be addressed. It would also be interesting to know if the authors have thoughts on how kinetochores may be moving directionally in the absence of Kip3. Is there an intrinsic mechanism that promotes kinetochore movement towards the microtubule plus-end? If so, has this been observed in previous in vitro studies of kinetochores?

We appreciate this comment and the opportunity to provide the following explanations and to modify our manuscript. In Figure 1, Mif2- and Cse4-labeled kinetochores move more slowly than kinetochores in which other proteins were tagged with fluorescent proteins. This observation is now addressed in the Results. These observations are consistent with the possibility that c-terminal GFP tags on Mif2 and Cse4 affect protein function and therefore dynamics on the microtubule. Both of the corresponding strains have slow growth rates. In addition, we had to make the linker for the GFP tag on Cse4 longer or else it was synthetic lethal with *cdc23-1* (doi: 10.1016/j.addr.2012.09.039). Despite the difference in measured lateral movement rates when these two kinetochore proteins were tagged, we believe that Figure 2 conclusively shows that Mif2- and Cse4- colocalize and move together with other kinetochore proteins.

With respect to Mtw1, while the speed was not significantly different from what was observed when kinetochore proteins other than Mif2 or Cse4 were tagged, kinetochores with tagged Mtw1 spend more time in minus-end directed movement when bound to the end of a microtubule than kinetochores in which other subunits were tagged. Whether this reflects a modification of Mtw1 function due to the fluorescent tag or some other factor whose nature is currently a mystery is unclear. However, the colocalization experiment in Figure 2 rules out the possibility that Mtw1 has a localization independent of other kinetochore proteins.

Taken together, our results indicate that the kinetics of microtubule-bound kinetochores in lysates is sensitive to perturbations in protein function and activity, but also that functional kinetochores all exhibit the same baseline characteristics of binding and plus-end directed movement on the lateral surface of microtubules followed by end-binding and tracking the depolymerizing ends of microtubules. We have modified the manuscript text to highlight these issues.

8. The data in Figure 4b appear to indicate that loss of cin8 activity increases the percentage of time that tip-bound kinetochores spend in minus-end directed movement. Is this a significant increase and, if so, is there a logical explanation for this effect? Cin8 has been proposed to promote disassembly of kinetochore microtubule plus-ends (PMID: 19041752).

We appreciate this comment. The data do indicate that in the absence of Cin8, end-bound kinetochores spend more time in minus-end directed movement. As highlighted in (Gardner et al., Cell 2008), Cin8 works with Kip1 as a microtubule crosslinker important for spindle integrity. Furthermore, their model for Cin8 promoting microtubule disassembly predicts a graded effect within the symmetry of the bipolar mitotic spindle. In contrast, our assay utilizes single microtubules grown from widely spaced seeds in order to achieve better resolution of individual molecular dynamics. Thus, our assay provides clearer observations of effects on single microtubules while sacrificing some of physiological geometry relevant to Cin8 function. Possible reasons why microtubules with end-bound kinetochores spend more time depolymerizing in Cin8 mutants include a change in protein composition at the kinetochore upon the loss of Cin8 or a more direct effect of Cin8 on kinetochore-microtubule dynamics. We prefer not to speculate on the biological implications of this observation in the absence of more data to form the basis for a model.

9. Please provide in the discussion a comparison of this study to the Tanaka re-activated CEN system (Tanaka et al., Nature 2005), which is still widely considered the gold standard in the field to directly observe different attachment configurations in yeast cells. In this system, no motor except Kar3 showed an effect. This should be addressed in the text.

We appreciate the opportunity to make this comparison. A new paragraph was added in the Discussion to frame our observations in the context of the Tanaka observations.. We believe that both mechanisms may work together to facilitate accurate chromosome congression and biorientation, making chromosome segregation robust to different cellular stresses.

10. Please provide a brief discussion regarding conceptual limitations of these lysates: Kip3 has roles in nucleus and cytoplasm, yet this distinction is lost in these whole cell lysates. This is especially critical for budding yeast which undergoes a closed mitosis. What are the prospects of using nuclear extracts instead?

We appreciate this suggestion and have added a discussion of the nuclear vs cytoplasmic compartmentalization to the Discussion section. The idea of using nuclear extracts or cytoplasmic extracts is very appealing to us. We have however not yet seen that achieving a clean separation of these fractions is possible for yeast, particularly for studies like ours that depend on minimizing lysate dilution.